# Nutrient limitation of algae and macrophytes in streams: Integrating laboratory bioassays, field experiments, and field data

Christopher A. Mebane[1]*, Andrew M. Ray[2], Amy M. Marcarelli[3]

1 Idaho Water Science Center, U.S. Geological Survey, Boise, Idaho, United States of America, 2 Greater Yellowstone Network, National Park Service, Bozeman, Montana, United States of America, 3 Department of Biological Sciences, Michigan Technological University, Houghton, Michigan, United States of America

* cmebane@usgs.gov

**Data Availability Statement:** The experimental data on which this article relies are available from a data repository (https://doi.org/10.5066/f72r3psj)

## Abstract

Successful eutrophication control strategies need to address the limiting nutrient. We conducted a battery of laboratory and in situ nutrient-limitation tests with waters collected from 9 streams in an agricultural region of the upper Snake River basin, Idaho, USA. Laboratory tests used the green alga *Raphidocelis subcapitata*, the macrophyte *Lemna minor* (duckweed) with native epiphytes, and in situ nutrient-limitation tests of periphyton were conducted with nutrient-diffusing substrates (NDS). In the duckweed/epiphyte test, P saturation occurred when concentrations reached about 100 μg/L. Chlorophyll *a* in epiphytic periphyton was stimulated at low P additions and by about 100 μg/L P, epiphytic periphyton chlorophyll *a* appeared to be P saturated. Both duckweed and epiphyte response patterns with total N were weaker but suggested a growth stimulation threshold for duckweed when total N concentrations exceeded about 300 μg/L and approached saturation at the highest N concentration tested, 1300 μg/L. Nutrient uptake by epiphytes and macrophytes removed up to 70 and 90% of the N and P, respectively. The green algae and the NDS nutrient-limitation test results were mostly congruent; N and P co-limitation was the most frequent result for both test series. Across all tests, when N:P molar ratios >30 (mass ratios >14), algae or macrophyte growth was P limited; N limitation was observed at N:P molar ratios up to 23 (mass ratios up to 10). A comparison of ambient periphyton chlorophyll *a* concentrations with chlorophyll *a* accrued on control artificial substrates in N-limited streams, suggests that total N concentrations associated with a periphyton chlorophyll *a* benchmark for desirable or undesirable conditions for recreation would be about 600 to 1000 μg/L total N, respectively. For P-limited streams, the corresponding benchmark concentrations were about 50 to 90 μg/L total P, respectively. Our approach of integrating controlled experiments and matched biomonitoring field surveys was cost effective and more informative than either approach alone.

and the field data are available from a data report (https://doi.org/10.3133/ds517).

**Funding:** Funding for the experimental components of this project was provided by the U.S. Environmental Protection Agency's Nutrient Criteria Program, through interagency agreement DW-14922442-01-0. Funding for the field data collection components of this project was provided by the U.S. Geological Survey's National Water Quality Assessment (NAWQA) program. No specific funding was received to write the article. The funders approved the general study design and encouraged publication of results but had no role in data collection and analysis, specific decisions to publish, or preparation of the manuscript. The manuscript was reviewed and approved for publication per U.S. Geological Survey Fundamental Science Practices.

**Competing interests:** The authors have declared that no competing interests exist.

## Introduction

Eutrophication due to nutrient enrichment and associated excessive growth of algae or aquatic plants is a well-known and persistent water quality concern across many agricultural or urbanized regions [1–3]. In the United States, nutrient management efforts are implemented to restore and maintain the chemical, physical, and biological integrity of surface waters in accordance with the Clean Water Act's total maximum daily load (TMDL) program. While these watershed plans are intended to define acceptable levels of pollutants, including nutrients, that are protective of beneficial uses, they commonly presume phosphorus (P) is the controlling nutrient and thus exclusively address nutrient management on P control [4]. This nutrient management presumption is common despite recommendations for restoring nutrient conditions in waters by evaluating the relationships among nutrients and algal response within stream systems with an understanding of which nutrient is limiting [5, 6]. In brief, nutrient limitation occurs when biological uptake of a macronutrient approaches its available supply. Operationally, we infer nutrient limitation when changes in growth or abundance of algae or aquatic plants are associated with changes in nutrient concentrations in surrounding waters. Nutrient saturation occurs when the availability of a nutrient increases to a point at which another factor critical to growth becomes limiting [7]. Understanding the limitation status of a waterbody is a necessary step for water managers, because it characterizes a waterbody's ability to retain nutrients exported from surrounding landscape [8, 9]. However, in broadscale analyses or in the establishment of numeric nutrient criteria, this step of empirically defining the limiting nutrient for an individual waterbody is often omitted. Instead, broadscale targets have been defined based on distributions of ambient data by assuming that the 25th percentile of historical data from all waterbody types approximates natural background conditions absent anthropogenic enrichment [10]. Likewise, broadscale empirical regression models that predict autotrophic biomass under varying nutrient concentrations frequently have low predictive power, especially from flowing waters. For example, Dodds et al. [11, 12] and Munn et al. [13] reported a number of regression models that predict benthic algal chlorophyll *a* as a function of nitrogen and/or phosphorus, and found $r^2$ values ranging from 0.03 to 0.44. The reasons for the weak relationship likely include the complex interactions of multiple physical and biological factors in stream environments [13–15] and that measured nutrient concentrations in water may not be a good proxy for the nutrient requirements at that time [e.g., 16].

The lack of simple patterns in field data of co-occurring nutrient and algae or plant abundance led to the desire to integrate the complex, community-level responses from field observations with single-species responses from laboratory experiments to see if together additional insights into nutrient-response patterns could be gained. The former provides the complexity of aquatic producer communities under ambient conditions while the latter offers confidence that response to nutrients can be examined without the confounding influence of environmental variability. Using a combination of independent lines of evidence offers advantages when deriving nutrient limitation information for establishing water quality plans. The approach here was to conduct controlled laboratory or instream manipulations of nutrient amounts and ratios in tests that were analogous to field measures. Specific goals and questions included:

1. To evaluate whether major nutrients, nitrogen or phosphorus, limited biological responses, either separately or together. If so, were nutrient ratios a useful way to estimate limitation?

2. To detect thresholds for accelerated growth of plants in response to increasing nutrient concentrations.

We investigated nutrient limitation in several streams in an agricultural region of the upper Snake River basin, Idaho, looking for commonality or contrast between three experimental

approaches. We also contrast our experimental results with numeric nutrient criteria guide-lines. Our experiments consisted of: (1) laboratory nutrient limitation experiments with green algae bottle tests using ambient stream waters and nutrient additions; (2) laboratory nutrient limitation experiments with the macrophyte *Lemna minor* (duckweed) and with epiphytic periphyton using ambient stream waters and nutrient additions; and (3) in-stream nutrient limitation experiments with nutrient-diffusing substrates naturally colonized by periphyton.

Bottle tests of algal productivity, sometimes called biostimulation assays or bioassays have long been used for measuring nutrient limitation of water [17, 18].These tests measure the response of a cultured species of green algae (Chlorophyta) in response to nitrogen (N) and P additions. While a major advantage of green algae bottle tests is their efficiency, their relevance for natural algal assemblies in streams is unclear. In seasonal monitoring of the streams that are the focus of the present paper, Chlorophyta were a minor component of periphyton abundance whereas Cyanophyta tended to be the most prevalent algae phylum followed by diatoms (Chrysophyta) [19]. In a mountain stream in Idaho, Marcarelli and Wurtsbaugh [20] found that while the relative abundances of different algal phyla changed with different nutrient availability, Chlorophyta were never dominant. Thus, the relevance of bottle tests with green algae remains to be demonstrated. Moreover the species of algae used in bottle tests is typically found in planktonic assemblages in lakes, and therefore may have different uptake affinities for limiting nutrients than periphyton that grow attached on substrates [21], as are more commonly found in streams, as well as vascular plants that can take up nutrients from the water column or from sediment via roots or shoots [22–24].

Macrophytes commonly represent a large portion of the primary producer biomass and serve as a substratum for epiphytes in aquatic ecosystems. Nevertheless, aquatic vascular plants are seldom used in nutrient enrichment bioassays, in part because macrophytes commonly obtain nutrients from both roots and shoots, complicating the design and interpretation of tests [23]. Free-floating duckweeds have advantages in this regard, as these small, fast-growing plants must obtain all nutrients from the water column and can readily be cultured and tested in laboratory settings. Ray et al. [25] demonstrated the responsiveness of a laboratory bioassay using with wild-harvested duckweed, *Lemna minor* L., and attached epiphytes to differing nutrient conditions in water collected from streams with differing nutrient conditions.

In contrast to the green algae bottle test and duckweed/epiphyte test in which stream waters are brought to the laboratory and tested in static conditions more reminiscent of lake waters, nutrient-diffusing substrates (NDS) are a tool for identifying nutrient limitation in stream periphyton communities [9]. In brief, concentrated nutrients are dissolved in a media (most often agar) to allow slow release, which is enclosed in a permeable container (e.g., a clay pot) or in vials with diffusive fritted glass tops, which allow nutrients to slowly leach through the substrates. Periphyton accumulates on diffusive surfaces when NDS are deployed in natural waters, and differences in accumulation rates between nutrient treatments and controls are interpreted as evidence of periphyton nutrient limitation. NDS have been used for approximately 40 years to determine nutrient limitation in lakes, streams, and large rivers [9, 26, 27].

These three testing approaches (single-species phytoplanktonic green algae in bottles, macrophyte/epiphyte growth in aquaria, NDS in streams) all have obvious differences from natural aquatic plant communities with their complex of bacterial, algal, and macrophytic assemblages. The point of this paper is to evaluate management tools for pragmatic tests of nutrient limitation and management, not to mimic ecological processes. Here we evaluate stream waters with phytoplankton, periphyton, and macrophyte tests and compare the experimental results to field measurements in the streams.

## Methods and materials

We investigated planktonic algae, benthic algae and macrophyte responses to nutrient gradients in experimental conditions that roughly corresponded with field measurements of the same concentrations. The studies began with the simplest tests, a single species of algae which was tested with water samples from selected field sites and increased in complexity to combined epiphytic periphyton and duckweed and in situ periphyton tests (Table 1).

The waters used for testing the responses to nutrient gradients were from selected sites surveyed in a broader field survey [19, 28]. A subset of these streams with a broad range of nutrient concentrations and ratios were further tested with nutrient additions. The selected experimental waters included reference streams with low nutrients but with differing ratios of nitrogen (N) and phosphorus (P) concentrations, streams that were slightly enriched above background, and highly enriched streams. Depending on the complexity of the experiments, between two and eight stream waters were tested. No regulatory permits were required for the collection of field samples reported herein.

We particularly focused on two test streams to evaluate P and N responses, one with very low N concentrations and one with very low P concentrations. The stream with very low N, Big Cottonwood Creek near Oakley, in south central Idaho represents a rangeland reference stream in near pristine condition. The stream is almost entirely in wilderness conditions upstream of the sampled reach with no diversions or channel alterations present and no livestock grazing. The Big Cottonwood Creek watershed is almost entirely roadless and the few roads that cross the watershed divide are located high in the watershed far from the stream, and no motorized off-road travel is permitted. While reference streams free from any of these disturbances are commonplace in higher-elevation, forested ecoregions, these types of human uses or disturbances are nearly ubiquitous in lower elevation, desert rangeland ecoregions [30]. Our field sampling showed total N to be very low (35–342 μg/L N) and total P concentrations averaged 35 μg/L, ranging from 18 to 66 μg/L over the course of the study. This allowed us to explore algal and macrophyte responses to enrichment in a low nitrogen stream under natural conditions.

The second stream that we focused on represented a situation with very low P concentrations (8–21 μg/L as total P) but ample nitrogen (1107 to 1643 μg/L N). This combination allowed us to design experiments to test for response thresholds with P, without the complicating role of potential nitrogen limitation or co-limitation. This stream, Stalker Creek, is located entirely within The Nature Conservancy's Silver Creek Preserve, and is protected to preserve its ecological services as part of a vulnerable high-desert spring-fed creek ecosystem.

These two streams were included in all aspects of the experiments listed in Table 1. For the planktonic green algae test and the in situ periphyton growth tests, other sites were also

**Table 1. Biological response variables measured in the field surveys and corresponding response variables tested in laboratory and instream nutrient limitation assays.**

| Variables that may describe eutrophication or biological condition | Field measurement | Relevant laboratory or in situ nutrient limitation experiment | Experimental endpoints |
|---|---|---|---|
| Sestonic algae abundance | Chlorophyll *a* concentration | Green algae growth test | Algal density at peak growth |
| Benthic algae abundance | Periphyton chlorophyll *a* and biomass | Nutrient-diffusing substrates (NDS) | Periphyton chlorophyll *a* and biomass on control and nutrient enriched artificial substrates, autotrophic index |
| | | Duckweed and epiphyte growth test | Periphyton chlorophyll *a* and biomass from different nutrient treatments |
| Macrophyte abundance | Biomass | Duckweed and epiphyte growth test | Duckweed biomass, plant size and abundance |

sampled that represented a range of nutrient concentrations. The experiments were also coordinated with field sampling of the same sites. Following a broader sampling of nutrient conditions in tributaries of the upper Snake River sampling in which 30 sites were sampled for a gradient of spatial conditions [28], in the second season a subset of sites were selected for repeated sampling to describe seasonal patterns in nutrient, algae, and macrophyte patterns [19]. Thus, with focused repeated field sampling and the nutrient-limitation tests described here, these latter streams represented a well characterized, rich data set to contrast with experimental manipulations.

## Green algae nutrient-limitation tests

The test organism was *Raphidocelis subcapitata (= Pseudokirchneriella subcapitata = Selenastrum capricornutum)* [31] from an axenic culture maintained by Aquatic BioSystems, Fort Collins, Colorado. The green algae population was incubated in the unamended ambient water samples in a static system for a common time period of at least 96 hours or until the maximum standing crop was achieved. The response of the population was measured in terms of changes in cell density, biomass, chlorophyll *a* content, or absorbance.

Waters from six sites (Table 2) were collected, kept chilled, shipped by overnight delivery to a testing laboratory, GEI Consultants, Littleton, Colorado. Upon receipt at the testing laboratory, samples were filtered through a 0.45 μm filter to remove any particulates or organisms that might interfere with the test. Four tests, with 3 replicates each, were conducted on each of six stream water samples, as follows: Site water only–baseline control (C treatment); Site water, plus 1 mg/L of N as $NaNO_3$ (N treatment); Site water, plus 0.1 mg/L of phosphorus as $K_2HPO_4$ (P treatment); Site water, plus both 1 mg/L N and 0.1 mg/L of P (NP treatment).

In order to estimate when maximum standing crop was achieved and thus when to end the test, absorbance readings were taken on days 3, 5, 7, 10, 12, and 14 of the test to determine when the percent change in biomass was less than 5% per day. When the change in biomass was less than 5% per day, maximum standing crop was assumed to have been achieved and the test was ended. This resulted in 12 to 14-day test durations. At test termination samples were filtered and dry weight was determined and used as the final endpoint of the test.

**Table 2. Summary of seasonal nutrient characteristics of streams and stream waters tested.** Streams are listed in order of increasing average total phosphorus (P) concentrations.

| Stream | Green algae | Duckweed/ epiphyte | Nutrient-diffusing substrates | Annual average (range) total N (μg/L) | Annual average (range) total P (μg/L) | n (nutrient samples) | Study site No. |
|---|---|---|---|---|---|---|---|
| Big Wood River | | | ✓ | 207 (50–560) | 9 (5.3–198) | 11 | 25 |
| Stalker Creek | ✓ | ✓ | ✓ | 1283 (1107–1643) | 12 (8.2–21) | 11 | 29 |
| Little Wood River | | | ✓ | 174 (93–407) | 30 (10–74) | 9 | 31 |
| Willow Creek | ✓ | | Ruined | 384 (245–620) | 30 (17–50) | 11 | 26 |
| Big Cottonwood Creek | ✓ | ✓ | ✓ | 139 (35–342) | 35 (18–66) | 11 | 10 |
| Camas Creek | | | ✓ | 2338 (502–4107) | 48 (21–147) | 11 | 28 |
| Goose Creek | ✓ | | ✓ | 400 (213–879) | 86 (30–238) | 11 | 8 |
| Billingsley Creek | ✓ | | ✓ | 1692 (1571–1816) | 93 (76–108) | 11 | 24 |
| Mud Creek | ✓ | | | 2889 (2883–2914) | 115 (103–127) | 2 | 17 |

Study site numbers are from Mebane et al. [28] and correspond with the KML map in the S1 File. Nutrient values were sampled across seasons in 2008 [19], except Mud Creek for which sampling was curtailed. Total N (organic nitrogen + ammonium + nitrate+ nitrite) and total P (all forms) were determined photometrically on unfiltered samples following alkaline persulfate digestion as described by Patton and Kryskalla [29].

Nutrient limitation was interpreted through 95th percentile confidence intervals (CI) and 2-factor analysis of variance (ANOVA). Confidence intervals are related to $p$ values, as any value outside the 95th percentile CI, when considered as a null hypothesis, gives two-tailed $p < 0.05$ [32]. Interpretation of differing outcomes of the 2-factor analysis of variance (ANOVA) followed Tank et al. [9]. Single nutrient limitation is indicated when just one of the nutrients (N or P) elicits a positive response, as determined by the mean response to the N or P treatment falling outside the upper CI of the control and the interaction term in the ANOVA is not significant. If neither N nor P alone significantly increases algal biomass, but N and P added together (N + P) do (that is, the interaction term in the ANOVA is significant; $p < 0.05$), then the algal biofilm is considered to be co-limited by both N and P. Secondary limitation is indicated if N or P alone significantly increases algal biomass, both N and P added together result in an even greater increase in biomass, and the interaction term for the ANOVA is significant. In such case, the nutrient that added alone produced a significant increase is considered the primary limiting nutrient, and the other nutrient is considered to be secondarily limiting [9].

## Macrophyte *Lemna minor* (duckweed) and epiphytic periphyton nutrient-limitation tests

The second experiment evaluated a more complex plant and a community algal response instead of just a single sestonic-algal species. Although aquatic vascular plants are seldom used in nutrient enrichment bioassays, macrophytes commonly represent a large portion of the primary producer biomass and serve as a substratum for epiphytes in aquatic ecosystems. Here, we used a laboratory bioassay designed with wild-harvested *Lemna minor* and attached epiphytes to integrate responses of the resident aquatic plant community to nutrient enrichment.

Methods followed the approach of Ray et al. [25]. In short, about 90L of water were collected from streams with low N and low P (Big Cottonwood and Stalker Creeks) and were transported to the Stream Ecology Center at Idaho State University, Pocatello, Idaho. *Lemna minor* and attached epiphytes were collected from a single location in the Portneuf River, Idaho. The low N waters were amended with nitrogen to determine if they were indeed nitrogen limited and if so, whether a response threshold could be detected. Ambient conditions plus four increasing N treatments were used, with three replicate aquaria for each treatment. The low P waters were similarly tested with an increasing P series. The 30 aquaria for both series, 5 treatments with 3 replicates each, were randomized on racks in a growth chamber. Maximum illumination intensity for the tests was about 5,500 lumens/m$^2$ and temperatures in the aquaria ranged from about 15.5°C in the dark to 19.0°C at peak illumination. Tests were concluded at 11 days. Water samples were collected at the start and end of the tests, filtered at 0.45 μM, chilled and acidified with $H_2SO_4$ to pH <2, and analyzed for total P and total N following alkaline persulfate digestion [29]. Response endpoints included duckweed biomass, root and frond lengths, numbers of plants and "benthic" chlorophyll *a* that was introduced as epiphytic periphyton with the duckweed and that then became established on the aquaria sides (S1 File). Ten plants per aquarium were randomly selected for root and frond measurements; periphyton chlorophyll *a* originating from epiphytes was collected by scrubbing aquaria walls, and the liberated periphyton was collected on filters for analyses, as described in [25]. For endpoints which increased with nutrient additions, effects concentrations associated with percentile increases were estimated by nonlinear curve fitting using OriginPro software (OriginLab, Northampton, Massachusetts).

### In-stream benthic algae nutrient limitation experiments with nutrient-diffusing substrates (NDS)

In this experiment we used in situ nutrient-diffusing substrates (NDS) to test for nutrient limitation of benthic algae. Our design used plastic tubes filled with agar with semi-permeable fritted glass discs for colonization surfaces held in place by heavy-gauge aluminum racks designed to withstand deployment in fast-water locations [33]. The design accommodated four treatments with six replicates each. Similar to the algal growth bioassay tests, each test included a control treatment with no nutrient amendments (C), a N-amended treatment, a P-amended treatment, and a N and P amended treatment (Fig 1). NDS were constructed by amending 2% agar with 0.5 mol $NaNO_3$ (N treatments) or 0.2 mol $KH_2PO_4$ (P treatments) or both (NP treatments). Amendments were added after the agar was removed from the heat source and had begun to cool. Amended agar solutions were poured into 35-mL polystyrene vials, which were capped with 2.6-cm-diameter porous, fritted porcelain crucible covers (Leco Corporation, St. Joseph, Michigan).

The NDS test sites were selected to provide a range of nutrient concentrations and ratios (Table 2). Racks with NDS were secured in locations with velocities and light that were reasonably representative for the study site and were incubated for 21 days. This incubation period was selected because it was just short of accrual periods that lead to sloughing and loss of benthic algae biomass, short enough to avoid extensive invasion of the racks by grazing snails, and because of experience that the nutrients in the vials become depleted by about 28 days [34]. Racks were secured with about 0.2 to 0.3m water depth above the growth discs. The racks were checked on days 11 or 12 to remove debris and make sure that they were not in danger of coming out of the water as flows dropped in late summer. Ambient nutrient samples were collected from stream water before and after the deployments and once during the deployments. Nutrient limitation was inferred using the same statistical approach as with the green algae tests.

## Results

Complete data for all of the tests are available from the ScienceBase.gov data archive [35].

### Green algae tests

Three of six stream samples (Mud Creek, Stalker Creek, and Willow Creek) showed primary P limitation, three showed N+P co-limitation (Stalker Creek, Willow Creek, and Big Cottonwood Creek). Only one sample (Goose Creek) showed primary N limitation (Fig 2).

By examining the pattern from all the streams with just the growth from the ambient water samples (no manipulations) we see an asymptotic relation with the curve starting to flatten at around 100 μg/L (~0.1 mg/L) total P, suggesting saturation. An asymptotic curve provided a reasonable fit to the data, suggesting a total P half-saturation value (50% of maximum) of about 35 ug/L (Fig 3). From the range of ambient P concentrations evaluated in these stream samples, there was no evidence of a low P threshold below which no or little growth response occurred in response to slight increases in P.

### Duckweed/epiphyte tests

The N series with Big Cottonwood Creek source water achieved a measured range of initial N concentrations ranging from 290 to 1277 μg/L with initial measured total P concentrations of 28 μg/L in all treatments. The P series with source water from Stalker Creek achieved a measured range of initial P concentrations ranging from 8 to 208 μg/L with initial measured N concentrations of 1277 μg/L in all treatments (Table 3).

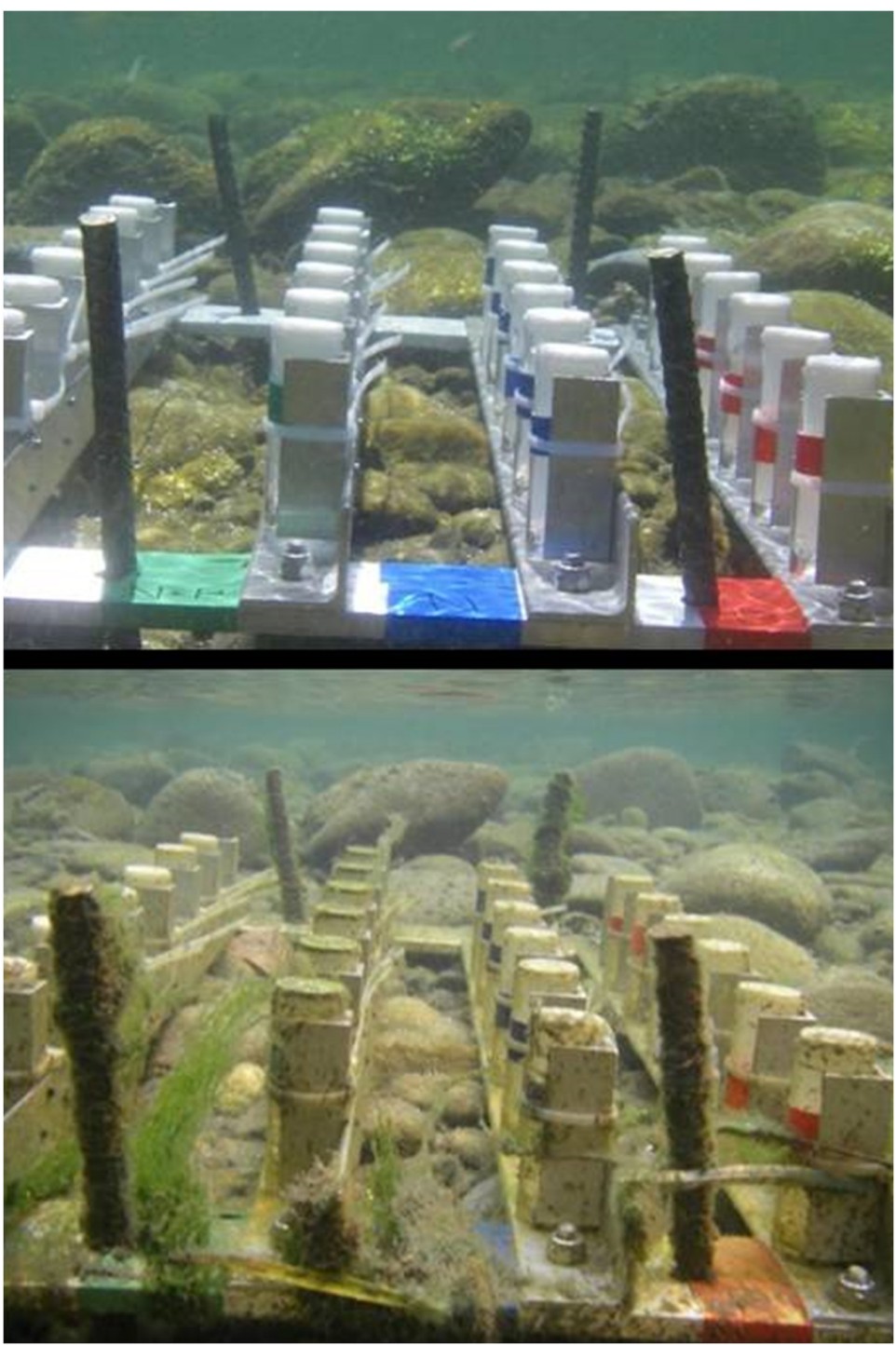

**Fig 1. Matched views of a nutrient-diffusing substrate (NDS) rack at deployment and after 21-days of colonization in a nitrogen and phosphorus (N+P) co-limited stream, the Big Wood River.**

Test results from the P series showed that duckweed biomass had a stepped pattern of increase with increasing P (Fig 4A). No substantial increase occurred until a threshold of about 50 μg/L and then no further increases were apparent beyond an apparent saturation point at around 100 μg/L P. The initial P concentration producing a 50% increase in biomass

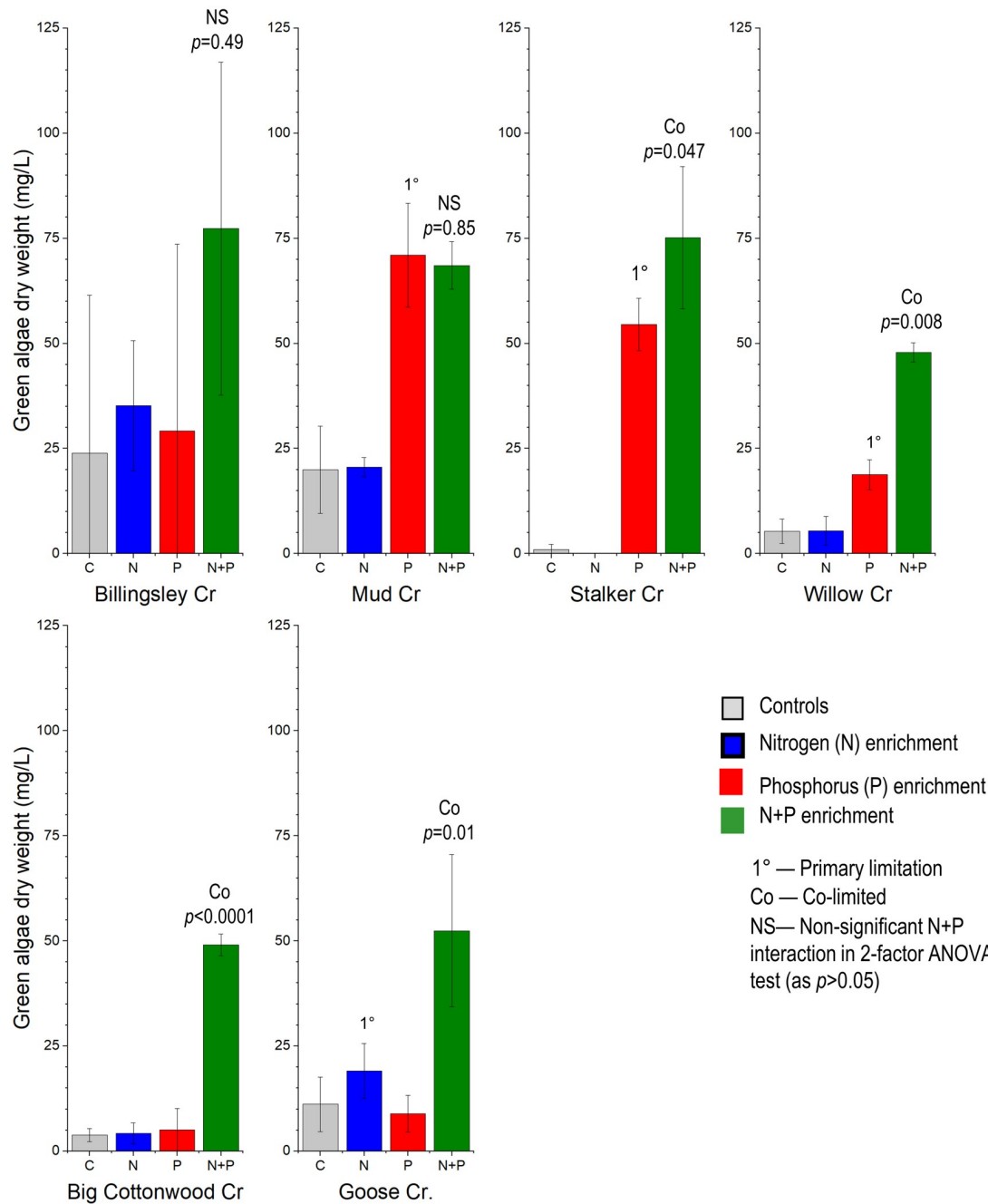

**Fig 2. Green algae growth results from six streams with Nitrogen (N), Phosphorus (P), or N+P additions.** Averages ± 95th percentile confidence intervals (CI) of the mean (n = 3). Primary nutrient limitation is concluded when the CI of the single nutrient additions does not overlap the mean of the controls. Co-limitation of N+P is determined by significant ($p<0.05$) interaction term in 2-factor ANOVA test.

(the EC50), estimated through logistic regression, was about 75 μg/L. In contrast, the periphyton had no threshold for initial response, but an exponential growth pattern up to about 75 μg/L with no further increases at higher P concentrations. Because at the 75 μg/L P treatment the N:P ratio was about 36, nitrogen should not yet have been limiting, suggesting a P saturation response (Fig 4).

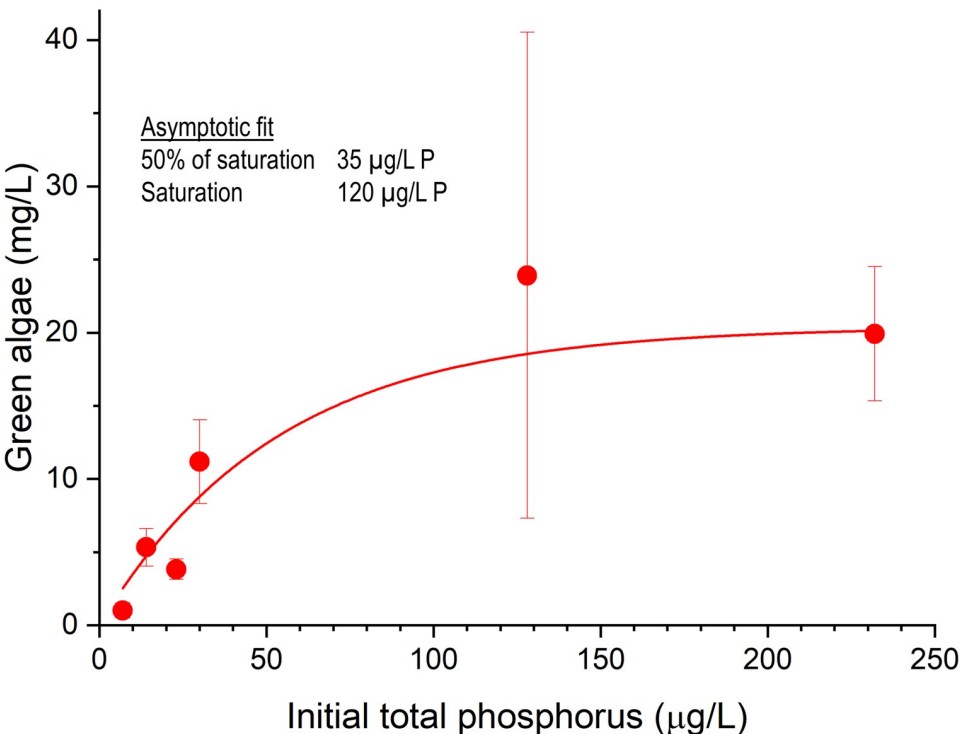

**Fig 3. Green algae growth in ambient waters (no additions).** The asymptotic relation with total P suggests saturation with a half-saturation value (50% of maximum) of about 65 μg/L TP and above about 120 μg/L TP, no further growth increases with P were noted. Error bars show standard deviation.

**Table 3. Percent phosphorus (P) and nitrogen (N) removed from the water column during the 11-day P and N growth series.**

**Phosphorus enrichment experiment**

| Initial P (μg/L) | Ending P (μg/L) | Initial N (μg/L) | Ending N (μg/L) | P % Removed | N % removed | Initial N:P molar ratios | Ending N:P molar ratios |
|---|---|---|---|---|---|---|---|
| 8.8 | 4.9 | 1277 | 1221 | 44% | 4% | 321 | 551 |
| 28 | 17.0 | 1277 | 1140 | 40% | 11% | 100 | 148 |
| 47 | 14.9 | 1277 | 440 | 68% | 66% | 61 | 65 |
| 80 | 10.6 | 1277 | 288 | 87% | 77% | 36 | 60 |
| 109 | 16.5 | 1277 | 458 | 85% | 64% | 26 | 61 |
| 208 | 23.7 | 1277 | 330 | 89% | 74% | 14 | 31 |

**Nitrogen enrichment experiment**

| Initial N (μg/L) | Ending N (μg/L) | Initial P (μg/L) | Ending P (μg/L) | N % removed | P % Removed | Initial N:P molar ratios | Ending N:P molar ratios |
|---|---|---|---|---|---|---|---|
| 290 | 264 | 28.2 | 16.8 | 9% | 40% | 23 | 35 |
| 319 | 240 | 28.2 | 11.0 | 25% | 61% | 25 | 48 |
| 434 | 210 | 28.2 | 10.2 | 52% | 64% | 34 | 46 |
| 708 | 467 | 28.2 | 9.8 | 34% | 65% | 56 | 105 |
| 1277 | 1140 | 28.2 | 17.0 | 11% | 40% | 100 | 148 |

Total N (organic nitrogen + ammonium + nitrate+ nitrite) and total P (all forms) were determined photometrically on 0.45 μM filtered samples following alkaline persulfate digestion as described by Patton and Kryskalla [29].

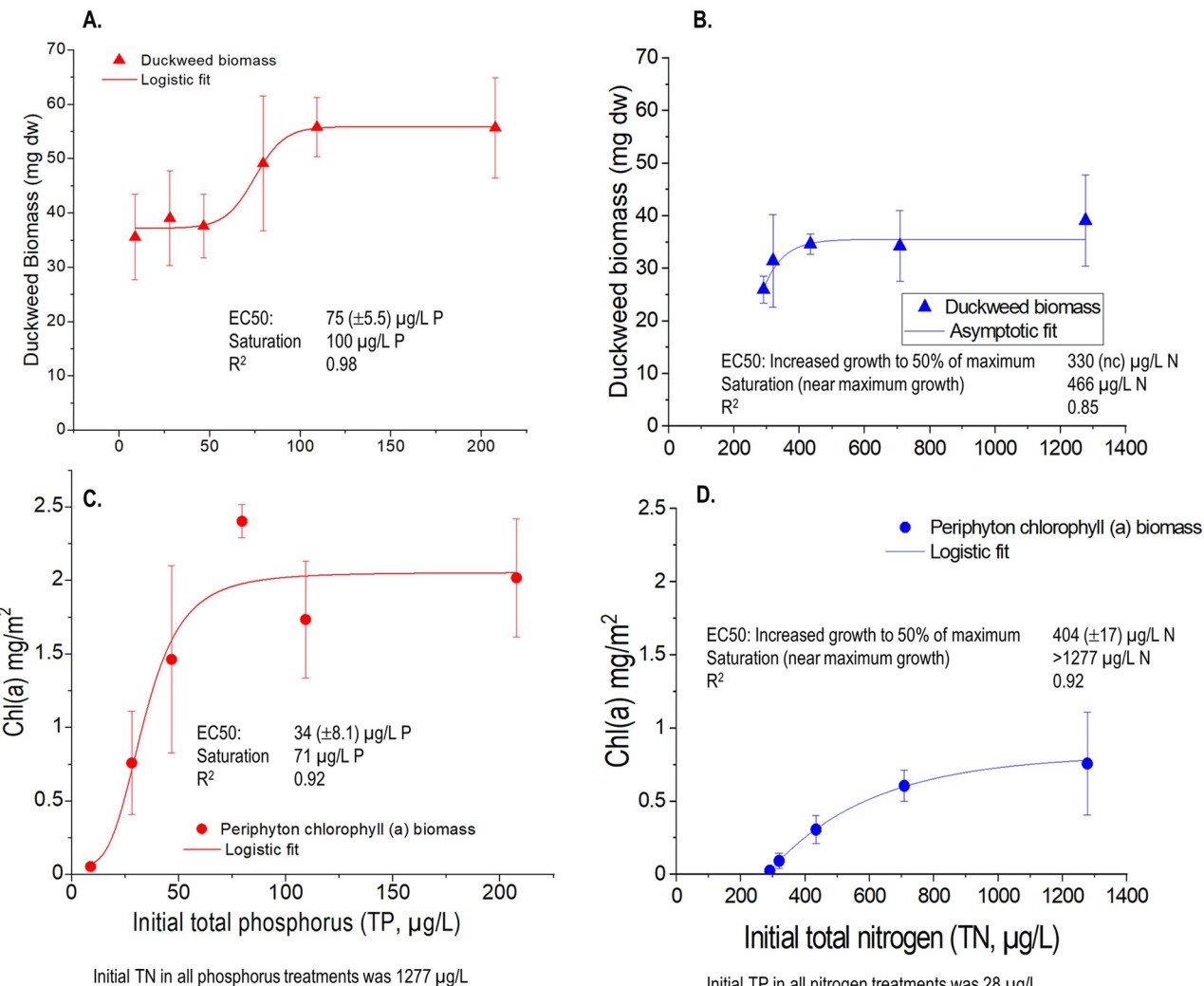

**Fig 4. Duckweed and periphyton responses to Phosphorus (P) enrichment (left) and Nitrogen (N) enrichment (right).** Error bars show standard deviations. EC50 (± SE): the concentration causing an increase in growth to 50% of maximum growth; saturation concentrations–further increases in P or N result in little further growth, calculated as 95% of the absolute asymptote.

The N enrichment series showed much lower plant growth than did the P series. Duckweed biomass showed only moderate increases with increasing nitrogen, suggesting that the abundant N had shifted the system into P limitation. The response of the algal community also showed no threshold of initial response, and although the absolute chlorophyll *a* biomass values were much lower than the P series, the shape of the growth curve was similar (Fig 4). By coincidence, the initial N concentration in the highest treatment in the N series was the same (1277 µg/L) as the N in the ambient water of the P series. This had the effect of providing a 6th and intermediate low treatment for the P series. This 28 µg/L point falls in line with the P growth series for both the algae and duckweed responses, lending further confidence in the reliability of the response curves.

Over the course of the experiment, up to 90% of P and nearly 80% of N were removed from the water column by the combined macrophyte and epiphyte growth. Even in the N series with low biomass accumulation, over 60% of P and 50% of N were removed (Table 3). This feature of static bioassays contributes to uncertainty in the interpretation of responses, curve

**Table 4. Nutrient limitation conclusions from the nutrient-diffusing substrate (NDS), average (ranges) of ambient nutrient concentrations in the streams at the time of the tests, and ancillary chemistry and channel measurements.**

| Stream | Stalker Creek | Big Cottonwood Creek | Big Wood River | Little Wood River | Goose Creek | Camas Creek | Billingsley Creek |
|---|---|---|---|---|---|---|---|
| Nutrient Scenario | Very low P; high N | Low P; Very-Low N | Very low P; very low N | Low P; very low N | Medium P; medium N | Medium P; very high N | High P; high N |
| NDS Nutrient Limitation | P+, Co | Co | Co | N+ | N+ | P+ | NP- |
| TN (µg/L) | 1329 (1133–1585) | 107 (35–151) | 100 (50–150) | 102 (93–118) | 302 (213–380) | 3446 (2505–3937) | 1706 (1571–1816) |
| $NO_3 + NO_2$ (µg/L as N) | 1233 (1024–1448) | <16 | <37 (<16–81) | 44 (15–76) | <16 | 3317 (2268–3944) | 1485 (1370–1600) |
| $NH_4$ (µg/L as N) | 13 (10–20) | <20 | <20 | <20 | <20 | <20 (<20–20) | 187 (170–200) |
| TP (µg/L) | 9 (8–10) | 36 (33–37) | 9 (7–10) | 13 (10–14) | 39 (30–43) | 38 (30–45) | 89 (87–90) |
| OP (µg/L as P) | 6.5 (3.9–7.9) | 32 (28–39) | 5.3 (2.7–9) | 5.5 (3.7–7.3) | 20 (12–32) | 26 (16–32) | 60 (57–62) |
| N:P molar ratios | 320 (306–340) | 6 (2.3–9) | 25 (15–37) | 18 (14–20) | 17 (16–21) | 209 (123–291) | 42 (38–45) |
| DOC (mg/L) | 1.3 (1.2–1.3) | 1.8 (1.3–2.1) | 1.2 (1.1–1.2) | 0.9 (0.7–1.0) | 3.4 (2.7–4.0) | 1.2 (0.7–2.1) | 1.1 (0.8–1.3) |
| Unshaded (%) | 96 | 69 | 71 | 88 | 89 | 99 | 98 |
| Turbidity (NTU) | 1.8 (1–3.2) | 2.4 (1.0–4.4) | 6.5 (2.6–9.9) | 0.65 (0.3–1.0) | 4.7 (3.1–6.5) | 1.4 (0.8–1.9) | 0.87 (0.6–1.0) |
| Qmax (L/s) | 1,120 | 2110 | 43,330 | 14,780 | 4163 | 24,350 | 1330 |
| Q (L/s) | 584 (541–671) | 49 (19–108) | 784 (680–906) | 520 (337–807) | 132 (79–204) | 53 (48–57) | 720 (576–838) |
| Wetted width (m) | 22 (14–34) | 3.7 (2.4–6.1) | 18 (12–27) | 13 (9.9–14) | 5.9 (3.1–8.7) | 9.0 (6.5–11.4) | 6.9 (5.2–8.1) |
| DO (mg/L) | 8.4 (4.0–14.1) | 8.3 (7.4–8.9) | 8.0 (7.7–8.4) | 8.9 (7.7–9.9) | 7.5 (5.7–8.8) | 9.8 (8.4–10.9) | 8.1 (5.5–10) |
| pH | 8.1 (7.7–8.7) | 7.5 (7.3–8.4) | 7.7 (7.5–7.8) | 7.9 (7.7–8.1) | 8.1 (7.7–8.4) | 8.2 (7.9–8.5) | 8.0 (7.5–8.4) |
| T (˚C) | 17.1 (14.3–23.5) | 16.7 (13.7–19.4) | 18 (15–20.7) | 11.4 (8.2–15.5) | 22.6 (20.8–24.8) | 16.8 (13.4–21.9) | 17.2 (15.6–18.8) |

N, nitrogen; P, phosphorus; NP, nitrogen + phosphorus co-limitation; +, stimulation response; -, suppression response; NP, colimitation by N and P as indicated by significant interaction term in 2-factor ANOVA ($p<0.05$); TN, total N (unfiltered); TP total P (unfiltered); OP, orthophosphate (filtered); Unshaded, percent of the sky over the NDS rack unshaded by vegetation or topography by digitized Solar Pathfinder images; NTU, nephelometric turbidity units; DOC, dissolved organic carbon; Q, streamflow; Qmax, maximum streamflow within the year prior to sampling; T, temperature, n = 3 for water samples, with samples near the start, middle, and ending of the NDS deployments, except for DO, T, and pH for Stalker, Cottonwood, Goose, and Billingsley, which were from 3-day continuous measurements. Brightbill and Frankforter [39] and Mebane et al. [28] (S1 File) give more details on field data.

fitting, and effect concentrations. Our interpretations focus on the initial rather than average concentrations because the uptake of nutrients by algae is usually rapid, growth over time is nonlinear (initially faster) and lags uptake [e.g., 36–38]. Effects concentration estimates would be lower if average instead of initial treatment concentrations were used. Using the same non-linear response curve fitting, the EC50 estimates for growth increases using initial or average P or N concentrations respectively were as follows: duckweed with P additions (Fig 4A), 75 vs 43 µg/L; epiphytes with P additions (Fig 4C), 34 vs 25 µg/L; epiphyte with N additions (Fig 4D), 404 vs 367 µg/L. No good fit with duckweed growth and average N concentrations could be obtained.

## Nutrient-diffusing substrates

Eight NDS experiments were attempted and seven were completed successfully (Tables 2 and 4). The unsuccessful test was ruined when drifting debris lodged on the rack late in the test, shading parts of the P and NP treatments and reducing velocity in all treatments.

Primary P limitation of chlorophyll *a* was present for 2 streams, Stalker Creek and Camas Creek (Fig 5a and 5f). Both streams had relatively low P, abundant N, and molar N:P ratios >200 (Table 4).

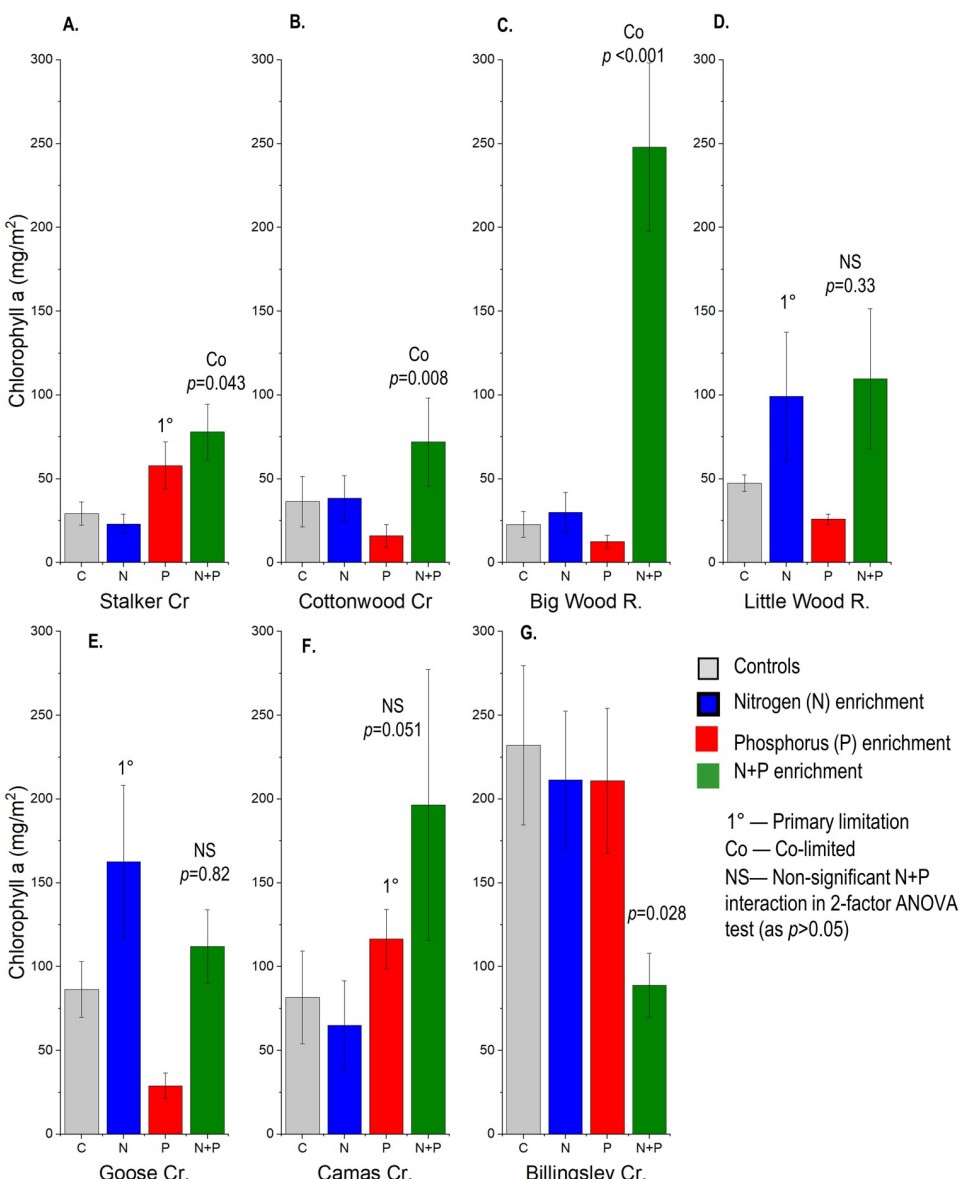

**Fig 5. Chlorophyll (a) accrual on nutrient-diffusing substrates (NDS) with Nitrogen (N), Phosphorus (P), or N+P additions.** Averages ± 95th percentile confidence intervals (CI) of the mean (n = 6). Nutrient limitation is interpreted as with Fig 2.

Primary N limitation was present for 2 streams, Little Wood River and Goose Creek (Fig 5D and 5E). Both streams had average N:P ratios of about 17–18, and very low to intermediate N concentrations (about 100 and 300 μg/L for Little Wood and Goose, respectively).

Co-limitation was present for at least 3 streams (Stalker Creek, Big Cottonwood Creek, and Big Wood River, Fig 5a–5c). Big Wood River had the lowest overall nutrient concentration of any stream in the study (P about 9 μg/L; N about 100 μg/L), followed by Big Cottonwood Creek with similar very low N concentrations but 4X higher P. Despite the high ambient N concentrations in Stalker Creek (~1300 μg/L), N was still co-limiting (Fig 5a). It is notable that Camas Creek with a N:P ratio >200, could also be co-limited as its P value of 0.051 for the interaction term was just above our arbitrary significance cutoff of P < 0.05 (Fig 5e).

Nutrient saturation of chlorophyll *a* was presumed for Billingsley Creek, a stream with high N and P and where none of the nutrient additions resulted in increased chlorophyll *a* accrual. Total P during the deployments was about 90 μg/L and N was about 1700 μg/L (Table 4).

Few clear patterns were observed between the nutrient treatments and total AFDM biomass. In Big Cottonwood Creek, AFDM clearly increased in response to N additions. As ranked absolute AFDM densities, N treatments were highest in all sites, but the responses were not statistically different using the CI overlapping the mean control response test, as used with the chlorophyll *a* responses (S1 Fig).

## Discussion

### Commonalities and contrasts in the nutrient-limitation tests

In both the green algae and NDS benthic algae tests, co-limitation by N+P was more common than single-nutrient limitation. The prevalence of co-limitation is consistent with many other studies of North American streams and broad syntheses of nutrient limitation in freshwaters [1, 33, 40–42]. This prevalence in turn supports the rationale for a dual-nutrient approach to eutrophication management, and the concern that a singular focus on P targets for 'TMDL' pollution reduction plans likely will not be fully successful for restoring or maintaining desired conditions of freshwaters [6]. An alternate perspective on eutrophication management advanced for N and P co-limited waters is that the focus should be on P controls [43]. However, advocates for a dual N and P management approach note that a singular focus on P reductions could inadvertently favor taxa that are commonly N-limited such as *Microcystis* or allow the export of elevated N downstream where algal production could be stimulated in lentic waters [44].

Conceptually, differences in nutrient requirements and thus limiting nutrients would be expected from a single-species green alga (Chlorophyta) test and complex benthic algae. However, for the 4 diverse streams tested with both methods, the conclusions on limiting nutrients were similar (Table 5). While not too much should be made of a comparison with a sample size of 4, this does suggest that comparatively simple green algae tests could have inferential value beyond sestonic green algae.

The lack of response from nutrients to Billingsley Creek with high N and P concentrations (~90 μg/L P and 1700 μg/L N) is indicative of nutrient saturation. This lack of response is also consistent with the duckweed and epiphyte test curves which had similarly flattened growth responses by these concentrations (Fig 4). Snyder et al. [45] conducted an NDS test in the nearby Snake River at King Hill, Idaho (their site S5) under lower N and P concentrations than in Billingsley Creek and also found nutrient-saturated conditions (62 μg/L P and 1360 μg/L $NO_3$ +$NO_2$ as N for Snake River vs. 90 and 1485), respectively, for Billingsley Creek (Table 4).

The single P additions tended to suppress algal growth as chlorophyll *a* at sites where N was low (Big Cottonwood Creek, Big Wood River, Little Wood River, and Goose Creek). In one instance (Billingsley Creek), N+P additions suppressed growth. Total periphyton biomass accrual as AFDM did not show this suppression pattern and was similar between control and

**Table 5. Nutrient limitation conclusions from streams tested under similar conditions with both the green algae bottle test and nutrient-diffusing substrates.**

| Stream | Stalker Creek | Big Cottonwood Creek | Goose Creek | Billingsley Creek |
|---|---|---|---|---|
| Nutrient Scenario | Very low P; high N | Low P; very-low N | Medium P; medium N | High P; high N |
| Green algae Nutrient Limitation | P, Co | Co | N, Co | Not nutrient limited |
| NDS Nutrient Limitation | P, Co | Co | N | Not nutrient limited |

P addition treatments, except for Big Cottonwood Creek where AFDM increased in response to P addition. In the green algae tests, the addition of more N to Stalker Creek, which already had a very high N:P ratio, killed the green algae (Fig 2).

Suppression of chlorophyll *a* growth in response to nutrient additions, particularly P, is a common phenomenon in nutrient limitation studies [9, 33, 46–48]. In Beck and Hall's [48] comprehensive review of the phenomenon, they found suppression in response to P in 13% of NDS studies reviewed, versus 4.7% and 3.6% for N or NP additions, respectively. The mechanisms for suppression in response to nutrient addition are unclear. Potential reasons include competition among the different algal species and heterotrophic organisms such as fungi and bacteria, direct toxicity resulting from mineral imbalance, or residual toxicity from $H_2O_2$ produced during autoclaving of agar when preparing the NDS treatments [48]. The latter, $H_2O_2$ toxicity, is clearly not relevant to our study, as we did not autoclave agar treatments. In each instance of the present study in which P suppression occurred, a strong positive response to N or N+P also occurred. To us, the fact that exacerbating N limitation or co-limitation by adding more P, supports the relevance of the NDS technique in assessing nutrient limitation in-stream.

## The use of N:P ratios to predict limiting nutrients

Nitrogen to phosphorus (N:P) ratios have long been used to predict potential nutrient limitation [49]. More recent studies have reached differing conclusions on the utility of N:P ratios for predicting nutrient limitation, ranging from support when overall nutrient concentrations are low [45] to useless [50]. The comparisons of N:P ratios and actual nutrient limitation determined in the present study give qualified support to the use of N:P ratios to predict transitions between N and P limitation, in waters with low overall nutrient concentrations. In instances where nutrient limitation was present and the total N:P molar ratios were >~30 (equivalent to mass ratios >~14), the algae or macrophyte growth was P limited or co-limited with N (Fig 6). However, probable N limitation was observed at N:P ratios up to 23 (equivalent to mass ratios up to 10), which is into the range for which P limitation is commonly predicted [>20, 49]). The Billingsley Creek results illustrate the point that N:P ratios lose relevance at high overall nutrient concentrations. Nutrient availability in this stream appears to exceed what could be used by algae, and the N:P ratio of about 40 was irrelevant for predicting limitation (Figs 2 and 5).

## The macrophyte conundrum

The stepped response of duckweed to P enrichment, with no increases in biomass in the low treatments was unexpected in a nutrient growth assay. The expected growth response is a steep initial response curve that flattens and approaches an asymptote as the nutrient additions approach saturation. The stepped response suggests competition with the attached epiphytes. Epiphytic algae coat the leaves and stems of macrophytes and may outcompete macrophyte for nutrients through faster uptake and physical exclusion [24, 28, 51].

Our water-only test results using duckweed biomass as a model macrophyte produced much higher effect concentrations than did water-only tests with pondweed, *Potamogeton pectinatus*, reported by Van Wijk [52]. Van Wijk [52] dosed *P. pectinatus* for 8-weeks in a gradient of P concentrations and found maximum growth had stabilized by 15 μg/L P and only about 7 μg/L P produced a 50% increase in growth (EC50) over the control treatment of 2.5 μg/L. In contrast, in our duckweed test, maximum growth stabilized at about 100 μg/L and 75 μg/L P produced a 50% growth increase. In Van Wijk's [52] study, the plants were kept free of epiphytes by regular and careful cleaning, whereas for our purposes, we wanted the

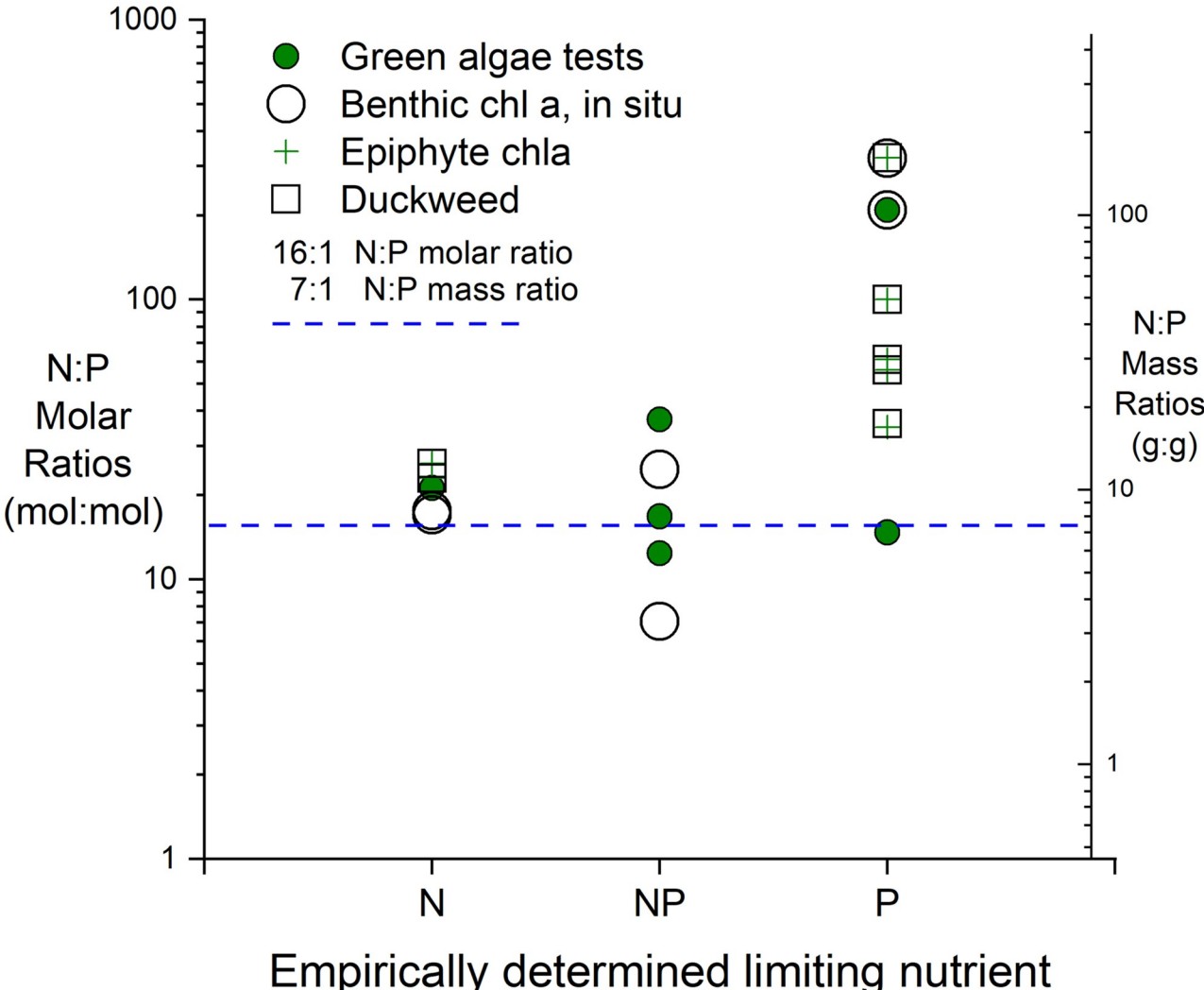

**Fig 6. Nitrogen to phosphorus ratios corresponding with empirically determined nutrient limitation from the different stream water experiments.** (N-nitrogen limited; NP-nitrogen and phosphorus co-limited; P-phosphorus limited).

additional complexity and increased realism of the epiphyte+macrophyte growth assay. We attribute the lack of growth response of the duckweed in the low P treatments below 50 µg/L P to competition from the epiphytes; however, whether this competition for P between the epiphytes and macrophyte could account for the large response concentration differences is uncertain.

Macrophytes may have strong influence on stream ecosystem structure and in excess can become a major nuisance. However, they are commonly excluded from nutrient enrichment testing or management strategies because of their complex life histories. We used duckweed as a convenient macrophyte for nutrient bioassays because its roots obtain nutrients from the water column rather than sediments. However, this makes it difficult to extrapolate to natural macrophyte beds. The protection or control of macrophyte assemblages by nutrient management in rivers and streams is confounded by the ability of some taxa to obtain nutrients through either sediments or the water column via roots or shoots, and the influence of factors such as water velocity, competition from epiphytic algae, or light limitation [23, 25, 28, 53].

The relative importance of sediment or water nutrient sources for rooted plants in streams probably differs by stream type and the taxa present. In streams that were subject to high velocities, macrophyte abundance was more strongly correlated with nutrients in sediment than water. In these streams, the taxa present were characterized by extensive and robust rooting, strong stems, and a streamlined morphology that bends with the current. Examples included *Stuckenia* sp., *Potamogeton* sp., and *Ranunculus* sp. [28]. In contrast, some macrophyte taxa found in lakes and slow-moving rivers may develop long stems, up to 2 m or more in length. In such cases, the root may function more as a holdfast rather than an efficient conduit of nutrients, and broken fragments can continue to survive, grow and re-attach. Examples of taxa that often develop long-stemmed forms include *Ceratophyllum* sp., *Elodea* sp., and *Myriophyllum* sp. [54, 55]. Further, macrophytes are shape shifters and the same species can take on very different morphological forms depending on hydraulic habitat and nutrient conditions. In oligotrophic conditions, some plants may develop robust roots and short-stemmed low profiles, versus weak rooted, long-stemmed profiles in eutrophic, low velocity settings [56, 57]. For instance, in different settings, *Potamogeton* sp. have been reported to both obtain nearly all their P from sediment [22, 23] or from water [24].

These complexities in turn make quantitative predictions of macrophyte abundance in streams in response to nutrient management complex and uncertain [e.g., 58]. Nevertheless, we think tests such as the duckweed/epiphyte exposure reported here can be helpful in both qualitative assessments and in contributing to the parametrization of numerical simulations.

## The role of nutrient uptake in the lack of simple correlations between nutrients and periphyton abundance

In addition to the nutrient limitation results, the magnitude of nutrient uptake that occurred in our macrophyte and epiphyte experiments with P and N suggests why correlations between ambient nutrients in water and algae abundance are often weak. For instance, a time-series plot of N, P, and benthic chlorophyll *a* in Big Cottonwood Creek over the two years of our study may simply appear chaotic upon initial glance (Fig 7a). Concentrations of N and P generally tracked together, but benthic chlorophyll *a* does not follow the same pattern. However, the green algae test, the macrophyte and epiphyte test, and the in situ benthic algal test (Figs 2 and 5b) all showed N limitation or NP co-limitation. These tests were conducted during both spring high-flow when N was near its annual maximum and summer base flows. Therefore, we focused on the N concentrations when comparing nutrients and benthic algae in these data. Removing P simplifies the plot of nutrients versus benthic algae, and it can more easily be seen that N and benthic chlorophyll *a* concentration vary in almost perfect synchrony (Fig 7c). However, N concentrations *decrease* as benthic chlorophyll *a* increases. While the low chlorophyll *a* value in May is likely attributable to scouring during high flows (S2 Fig), the pattern can be seen at other times of the year. These results are consistent with other work showing high potential for nutrient uptake in streams, linked to instream biotic activity [59–61]. Our macrophyte/epiphyte experiment is also consistent with the decline of N from the water column corresponding with increases in algal chlorophyll *a* in Fig 7. Even though the absolute plant biomass in the macrophyte/epiphyte experiments was lower than that measured instream, at times greater than 50% of the N in the microcosms was taken up by the plants and incorporated into biomass even over the relatively short duration of the tests (Table 3).

The rapid removal of nutrients from the test solutions suggests practical implications for monitoring and water management. Because the plant densities in our tests were low, representative of oligotrophic conditions, we think the magnitude of nutrient removal has important implications for interpreting monitoring data. Further, this supports the utility of using

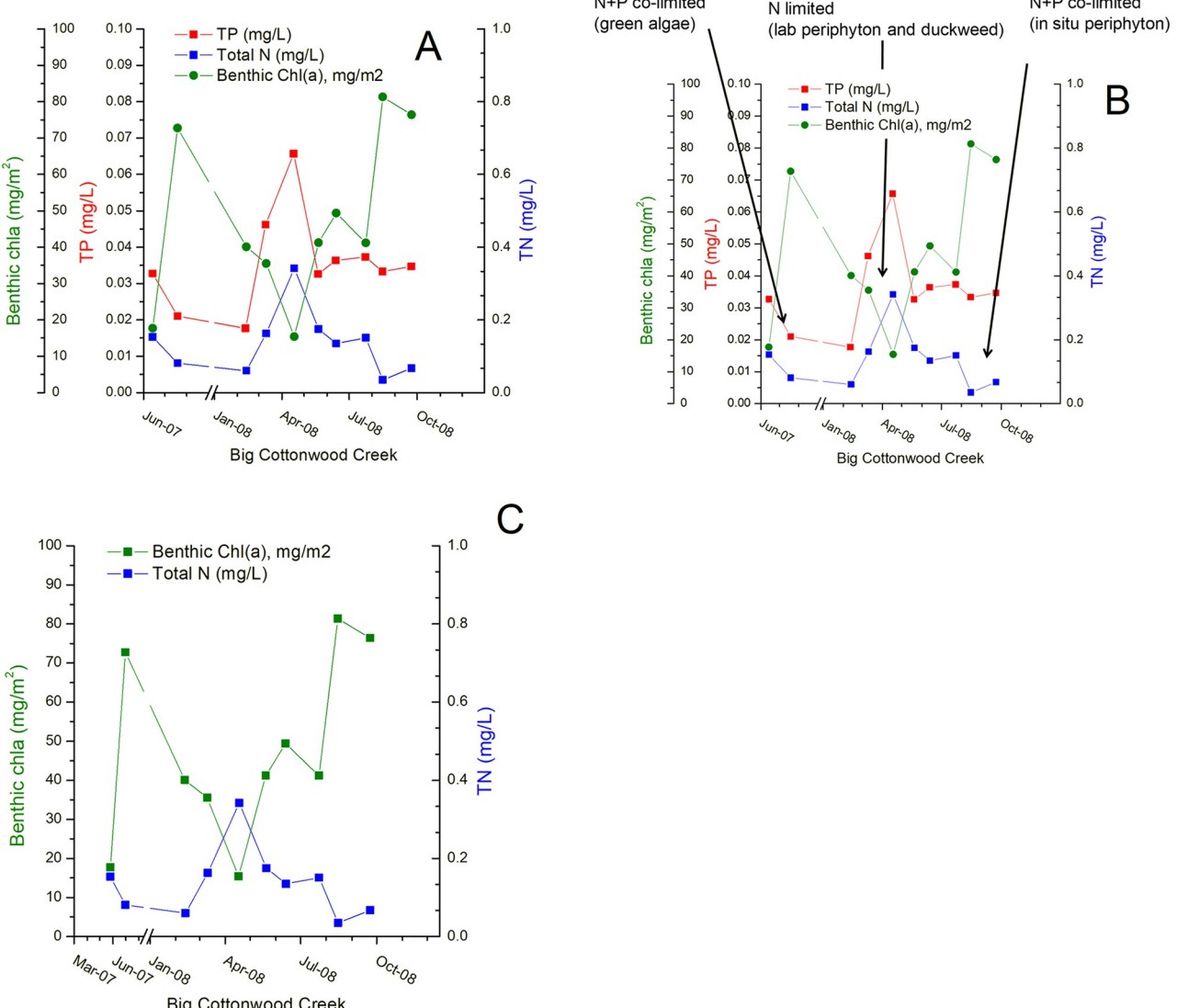

**Fig 7. Benthic chlorophyll *a* density over time in a nitrogen-limited stream.** A. Benthic chlorophyll a relative to both nitrogen (N) and phosphorus (P); B. Overlain with the three nutrient limitation experiments conducted on the stream that demonstrated consistent N limitation; C. A simplified comparison without P shows nearly perfect inverse correspondence between chlorophyll *a* and N.

duckweed as a low cost, effective means of removing nutrients from wastewater ponds, such as aquaculture settling ponds, municipal sewage or lagoons for confined animal feeding operations. This is likely a feasible approach in many places so long as pond residence times are at least few days and a skimmer to remove plants can be used.

## Predicting relations between nutrients and benthic chlorophyll *a* abundance

An important question is whether the benthic algae accrual on the artificial substrates can be used to predict nutrient-algae abundance relations in streams. First, when comparing the periphyton biomass that was measured on natural substrates at the time of the NDS artificial substrate experiments to that measured on the non-nutrient amended controls of the NDS

experiments, in some instances the biomasses were similar but more frequently the periphyton biomass was higher on the artificial substrates than on natural substrates from the same sites while the experiments were underway. This tendency for high growth on fresh, uncolonized substrates is probably related to the lack of competition and is similar to the high growth that sometimes occurs after freshets or other bed disturbances [e.g., 62]. Other researchers have found that colonization periods of about 30 days allow for maximum biomass accrual on artificial substrates, but avoids subsequent sloughing or invasion by snails or other grazers [45, 63, 64].

Second, benthic algal abundance on natural substrates in relation to nutrients were variable to the point of appearing chaotic and challenging to interpret even with intensive monitoring efforts. Yet, when benthic chlorophyll *a* biomass that had accrued after 21-days on the controls of the NDS artificial substrates was compared to N from those streams that were experimentally shown to be N limited, a consistent pattern was apparent (Fig 8). The curve shape from these ambient N concentrations and benthic chlorophyll *a* is unmistakably similar to the asymptotic, rate limited curve shapes seen in the green algae and epiphyte N and P series (Fig 4). Together, these two observations suggest that the relation of benthic chlorophyll *a* biomass and ambient N may approximate an upper limit for average chlorophyll *a* biomass in streams in relation to late summer N concentrations.

Overlaying the 62 matched benthic chlorophyll *a* and N samples collected during summer (June-Sep) in 2007 and 2008 for the field study gives a good fit between the N-chlorophyll *a* curve developed from colonizing artificial substrates in N-limited streams and the upper limit of chlorophyll *a* and ambient N concentrations (Fig 8). Because most sites were N or co-limited in the in situ tests, similar comparison with P are comparatively poor. Nevertheless, fitting a logarithmic curve to the 4 NDS sites that were either P limited or co-limited also produced a function that reasonably matched the upper limits of the chlorophyll *a* collected from natural substrates, plotted as a function of P (Fig 9).

An implication of these patterns is that they approximate the maximum chlorophyll *a* biomass in streams for given N or P concentrations. Below the curves, chlorophyll *a* is presumably prevented from reaching these maximum biomasses because of other limiting factors such as P limitation for the N curve, N limitation for the P curve, light limitation, grazing pressure, or a host of other measured or unmeasured factors. The limiting function inferred from overlaying the chlorophyll *a* accruals on fresh substrates in Figs 8 and 9 follow the upper edge of the cloud of points for N and P. This is conceptually similar to inferring ecological relationships from the edges of scatter plots by using quantile edge regression approaches (e.g., 80th or 90th percentile, depending on dataset size) [5, 65, 66]. While the Figs 8 and 9 scatter plots with lines overlying the edge of the points look similar to edge regression plots, the difference is that rather than fitting a line to the edge of the distribution of a dataset in edge regression, the curved lines in Figs 8 and 9 were from the independent NDS exposures.

Chlorophyll *a* values of 150 and 200 mg/m$^2$ have been considered to define desirable or nuisance conditions in rivers, respectively, based on public opinion surveys of recreationalists [67]. From the patterns shown in Fig 8, mean summer N concentrations of about 600 to 1000 µg/L in N-limited streams would be expected to limit periphyton chlorophyll *a* to about 150 to 200 mg/m$^2$. From Fig 9, mean summer P concentrations of about 50 to 90 µg/L in P-limited streams would be expected to limit periphyton chlorophyll *a* to about 150 to 200 mg/m$^2$ respectively.

We think that this approach and the specific associations between nutrient concentrations and periphyton densities in streams are an appreciable advance for estimating nutrient benchmarks ('criteria') to avoid undesirable plant growth over commonly used ecoregional distributional models (e.g., setting targets at the 25th percentile level of available data for an ecoregion)

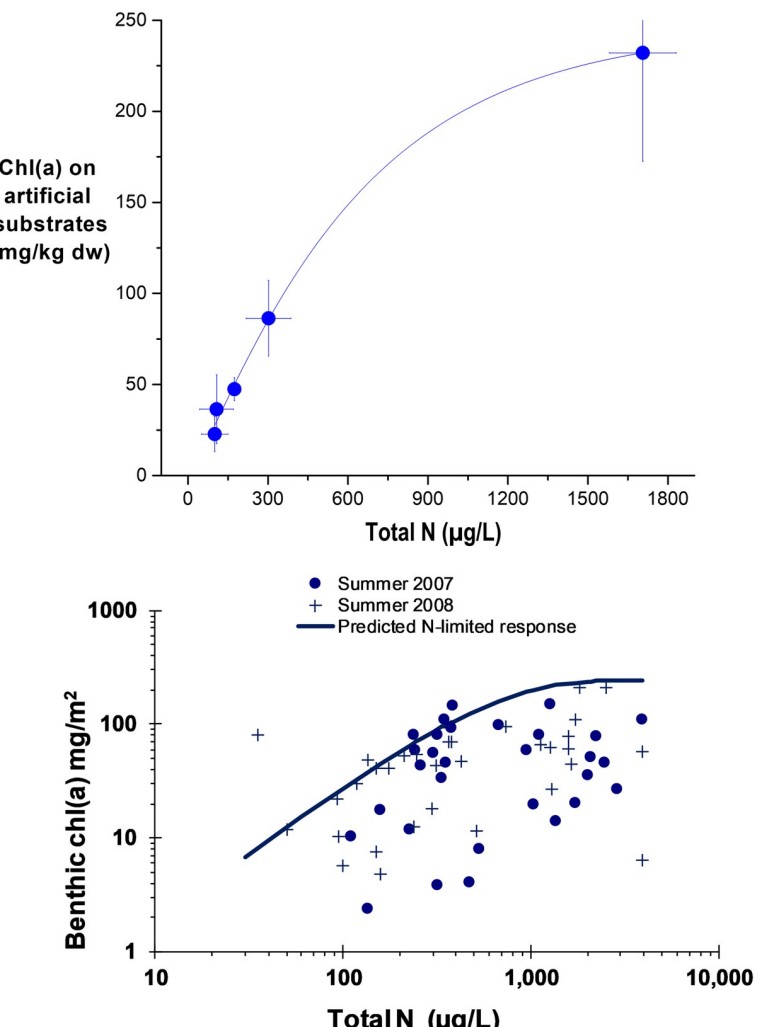

**Fig 8. Benthic chlorophyll *a* on artificial substrates at nitrogen (N) limited sites versus field samples.** Curve fit to chlorophyll a from NDS controls (no nutrient additions), excluding P-limited sites, following 21-day accrual (top), and the same line fit overlaying 62 benthic chlorophyll *a* samples collected from natural substrates in associated field collections (bottom). Field data from [28, 39].

[10]. However, there are important limitations to our study and cautions to applying the specific concentration values to waters that are greatly dissimilar to the study streams. Algal growth in the environment is not as simple as extrapolating lab curves, and other factors such as grazing, flow, light penetration, temperature, and sloughing could greatly influence or control algae biomass in streams and rivers [58, 68].

The nutrient benchmarks derived here from the NDS and field survey data are considerably higher than benchmarks derived from some artificial stream studies [69–71]. For instance, Schmidt et al. [70] showed P saturation occurred between about 12 and 29 µg/L P when N was not limiting and N saturation occurred between 150 to 2,450 µg/L when P was not limiting [70]. Phosphorus concentrations which saturate cellular level growth kinetics are very low, at less than 1 µg/L of soluble reactive P, yet in nature, much higher P levels are usually required to cause more dense accumulations of periphyton [69]. Bothwell showed that as benthic algal colonization proceeded, an initial saturation plateau controlled by cellular growth rate kinetics

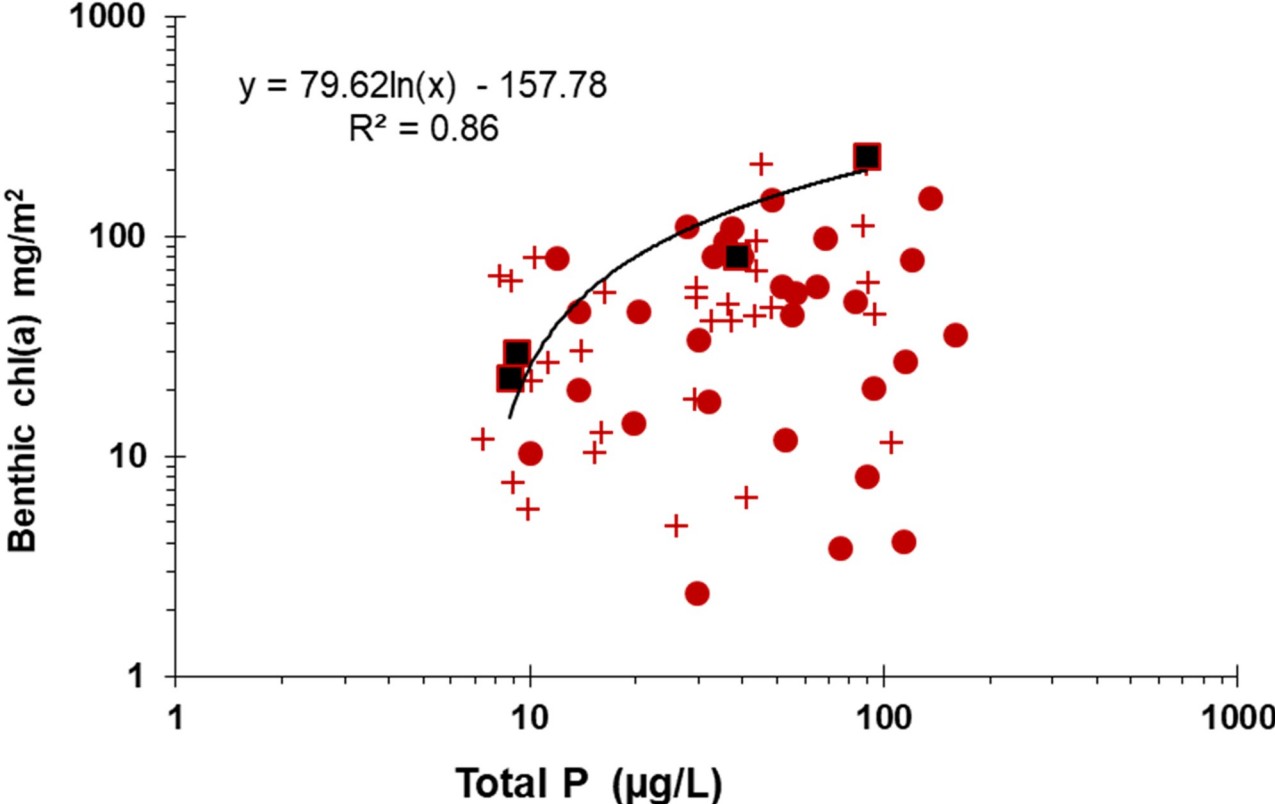

**Fig 9. Benthic chlorophyll *a* on artificial substrates at phosphorus (P) limited sites versus field samples.** Curve fit to chlorophyll a from NDS controls (no nutrient additions), excluding N-limited sites, following 21-day accruals overlain with 62 benthic chlorophyll *a* samples collected from natural substrates.

developed in the first two weeks at about 0.5 to 1 µg/L P, followed by slow increases limited by diffusion-limited kinetics up to an ultimate plateau of about 25 µg/L after about 30 days [69]. These changing saturation processes and >20 fold differences in P saturation concentrations caution that saturation thresholds derived from fresh benthic colonization laboratory assays may not be directly applicable to natural, established algal communities that develop throughout the growing season. For example, Schmidt et al. [70] designed their tests (short-term with strong current to overcome any diffusion gradients) to produce maximum growth to define rate constants in numerical models, where in application limiting factors such as layer diffusion, sloughing, and light limitation would be adjusted by other mathematical functions.

The nutrient benchmarks derived here are also considerably lower than some benchmarks derived for naturally turbid streams. The NDS deployments and field survey data used in this study are from shallow, fairly clear water streams with good light penetration. Nutrient benchmarks derived for turbid waters with limited light penetration have been considerably higher. For instance, Suplee et al. [72] used a process-based model to derive

nutrient criteria of 55 μg/L P and 655 μg/L N to limit benthic algae growth in the Yellowstone River upstream of a turbid tributary, the Powder River (Montana, USA). These values are effectively the same as our estimates for the streams in the present study. In the more turbid section of the Yellowstone River downstream of the Powder River, 95 μg/L P and 815 μg/L N were projected to produce the same benthic algae crop [72]. Chambers et al. [73] derived P and N criteria of 102 and 980 μg/L respectively for turbid, western prairie streams of Canada. In contrast, criteria for clearwater, montane streams in inland British Columbia were 20 and 210 μg/L, P and N respectively [73].

## Conclusions

Conclusions that we draw from this work include:

1. Different endpoints (e.g., phytoplankton algae, periphyton algae, and the macrophyte) tested in the same or similar waters sometimes had different limiting nutrients, however nitrogen (N) limitation or co-limitation was most common;

2. With both green algae and algae in periphyton, phosphorus (P) had no minimum response threshold. Rather, in both test series algae biomass followed an exponential growth function with increasing P concentrations up to an apparent saturation threshold of around 100 μg/L total P (TP) with no further growth increases at higher P concentrations;

3. With the macrophyte test, an apparent P threshold of response for increased growth was around 50 μg/L TP and an apparent saturation threshold was around 100 μg/L;

4. In the growth experiments with duckweed and epiphytic periphyton, most of the N and P in the test solutions was removed over the course of the 11-day tests. This suggests that uptake of N and P in oligotrophic streams likely complicates relations between plant biomass and nutrient concentrations in stream surveys;

5. A comparison of ambient periphyton chlorophyll *a* concentrations with concentrations accrued on control artificial substrates in N-limited streams, suggests that N concentrations in clearwater N-limited streams associated with periphyton chlorophyll *a* thresholds for desirable or undesirable "too green" conditions for recreation would be about 600 to 1000 μg/L, respectively;

6. A comparison of ambient periphyton chlorophyll *a* concentrations with concentrations accrued on control artificial substrates in P-limited streams, suggests that TP concentrations in clearwater P-limited streams associated with periphyton chlorophyll *a* thresholds for desirable or undesirable conditions for recreation would be about 50 to 90 μg/L, respectively; and,

7. Integrating controlled experiments and matched biomonitoring field surveys was more informative than either approach alone. Considering relatively low costs of testing such as we have described here versus engineering and construction costs of wastewater treatment (on the order of tens of thousands versus tens of millions or more dollars respectively) that might not address the correct limiting nutrient, we recommend wider application of integrated nutrient-limitation testing approaches.

## Supporting information

**S1 Fig. This figure showing (1) biomass (as AFDM) responses to nutrient enrichment and (2) interactive plots of time series nutrient, organic carbon, streamflow, benthic algae, and**

**macrophytes in the study streams.**
(PDF)

**S2 Fig. This figure showing (1) biomass (as AFDM) responses to nutrient enrichment and (2) interactive plots of time series nutrient, organic carbon, streamflow, benthic algae, and macrophytes in the study streams.**
(PDF)

**S1 File. This includes more details and photographs of the study sites.**
(PDF)

**S2 File. This shows site locations, accessible through spatial viewers such as QGIS or Google Earth.**
(KML)

**S1 Image.**
(JPG)

## Acknowledgments

The algal growth potential tests were directed by Suzanne Pargee, GEI Consultants, Littleton, CO. Flint Raben, Idaho State University Department of Biological Sciences, Pocatello, ID (ISU) supported the duckweed collection and testing and Kelsey Flandro, ISU, conducted the chlorophyll *a* and biomass measurements for both the *Lemna* and epiphyte growth tests and the NDS experiments. Diana Eignor, project officer with the U.S. Environmental Protection Agency's Nutrient Criteria Program, encouraged and facilitated this work. The views expressed in this article are those of the authors and do not necessarily represent the views or policies of the U.S. Environmental Protection Agency. Any use of trade, firm, or product names is for descriptive purposes only and does not imply endorsement by the U.S. Government.

## Author Contributions

**Conceptualization:** Christopher A. Mebane, Andrew M. Ray, Amy M. Marcarelli.

**Data curation:** Christopher A. Mebane.

**Formal analysis:** Christopher A. Mebane.

**Funding acquisition:** Christopher A. Mebane.

**Investigation:** Christopher A. Mebane, Andrew M. Ray, Amy M. Marcarelli.

**Methodology:** Christopher A. Mebane, Andrew M. Ray, Amy M. Marcarelli.

**Writing – original draft:** Christopher A. Mebane.

**Writing – review & editing:** Andrew M. Ray, Amy M. Marcarelli.

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
