## [Decision Letter · Decision Letter 0]

3 Feb 2021

PONE-D-20-39177

Nutrient limitation of algae and macrophytes in streams: integrating laboratory bioassays, field experiments, and field data

PLOS ONE

Dear Dr. Mebane,

Thank you for submitting your manuscript to PLOS ONE. After careful consideration, we feel that it has merit but does not fully meet PLOS ONE’s publication criteria as it currently stands. Therefore, we invite you to submit a revised version of the manuscript that addresses the points raised during the review process.

We look forward to receiving your revised manuscript.

Kind regards,

Frank Onderi Masese, Ph.D

Academic Editor

PLOS ONE

Journal Requirements:

2. In your Methods section, please provide additional location information of the study sites, including geographic coordinates for the data set if available.

3.We note that you have indicated that data from this study are available upon request. PLOS only allows data to be available upon request if there are legal or ethical restrictions on sharing data publicly. For more information on unacceptable data access restrictions, please see http://journals.plos.org/plosone/s/data-availability#loc-unacceptable-data-access-restrictions.

4.Thank you for stating the following in the Acknowledgments Section of your manuscript:

"Funding for the field component of this project was provided by the U.S. Geological Survey’s National Water Quality Assessment (NAWQA) program. Funding for the experimental components of this project was provided through a cooperative agreement with the U.S. Environmental Protection Agency’s Nutrient Criteria Program, Diana Eignor, project officer."

 "The funders approved the general study design and encouraged publication of results but had no role in data collection and analysis, specific decisions to publish, or preparation of the manuscript. The manuscript was separately reviewed and approved for publication per U.S. Geological Survey Fundamental Science Practices."

Additional Editor Comments:

I have now received review comments by two reviewers who agree on the strength of the paper and its contribution to nutrient management in streams and rivers. While both reviewers have provided specific and general comments on how the paper can be improved, reviewer 2 is very critical of the writing style used. Thus, the paper need reorganization using a parallel structure. There is also a need to shorten some sections of the paper as indicated.

Reviewers' comments:

Reviewer's Responses to Questions

**Comments to the Author**

1. Is the manuscript technically sound, and do the data support the conclusions?

Reviewer #1: Yes

Reviewer #2: Yes

2. Has the statistical analysis been performed appropriately and rigorously? 

Reviewer #1: No

Reviewer #2: Yes

3. Have the authors made all data underlying the findings in their manuscript fully available?

Reviewer #1: Yes

Reviewer #2: Yes

4. Is the manuscript presented in an intelligible fashion and written in standard English?

Reviewer #1: Yes

Reviewer #2: No

5. Review Comments to the Author

Reviewer #1: This is a nice paper of practical and ecological importance. The growth assays combined with the field measures makes the work stronger and helps us start to understand linkages between laboratory and field experiments commonly used to assess nutrient limitation in streams. Nutrient pollution in streams is an important management issue, so the work has practical implications as well.

My comments are mainly on details, not the overall scope or conclusions of the manuscript.

Line 152. Was the algal culture axenic?

Line 168. Given that multiple ANOVA tests were run, was there any effort to do Bonferonni correction or a giant combined test of results?

Line 189. What light intensity was used for the two types of bioassay experiments?

Line 206. Were the NDS agar solutions mixed with phosphorus at the last minute upon cooling? Otherwise inhibitory compounds can be produced (e.g. reference 8). While later on the authors state that the agar was not autoclaved, this still could happen with boiling. It really is the best explanation (in my mind) for the inhibitory response.

Line 233. A growth function as a function of nutrients is Monod, Michaelis-Menten is for nutrient uptake. Same mathematical form, but technically different. Line 264 makes that distinction.

Line 245. Stepped increase very interesting result

Line 253. Larger plants have much more capacity to be plastic in their internal nutrient content and can store nutrient up from prior pulses. Line 261 supports this hypothesis. Lohman and Priscu (1992) show this for Cladophora (kind of on the border between a macro and a macrophyte…)

Line 456. Good explanation for Figures 8 and 9 that indicate that nutrient bioassays show that initial burst of growth, but are and are not subject to other controls of natural standing stock. Maybe consider that a quantile regression is good for maximum chlorophyll? This approach has been taken previously (or using maximum chl rather than seasonal means from standing stocks)

Figure 2. I don’t think it is necessarily correct to calculate 95% confidence intervals when there is a significant interaction?

Figure 7 shows why longer term means are useful. For example figure a has a spike in TN and TP and a decline in benthic chlorophyll. This could be a sloughing event and the TN and TP made mostly of lost algae.

Figure 8 a. Not sure how you would get chlorophyll per kg of periphyton on the glass frits.

Lohman, K., and J. C. Priscu. 1992. PHYSIOLOGICAL INDICATORS OF NUTRIENT DEFICIENCY IN CLADOPHORA (CHLOROPHYTA) IN THE CLARK FORK OF THE COLUMBIA RIVER, MONTANA1. Journal of Phycology 28:443-448.

Reviewer #2: Review of PONE-D-20-39177 – “Nutrient limitation of algae and macrophytes in streams: integrating laboratory bioassays, field experiments, and field data” by Mebane, Ray and Marcarelli.

This manuscript presents the results from a study that used multiple laboratory- and field-based approaches to explore the boundaries of N and P limitation in a group of western US streams that lie along a gradient of anthropogenic nutrient enrichment. Nutrient limitation was assessed using a combination of low-concentration threshold and high-concentration saturation responses with bottle bioassays of a single algal taxon, laboratory growth trials of Lemna and its epiphytes, and deployments of nutrient-diffusing substrates in the field. This is an interesting combination of approaches that could inform the setting of nutrient criteria in streams. Perhaps not surprisingly however, the manuscript struggles to organize all the various components of the results in a clear and logical way. I have both general and specific suggestions for its further improvement.

General Comments

Abstract: The abstract suffers from a number of problems. First, the abstract seemingly uses the terms “limitation” and “saturation” interchangeably. Growth stimulation at low concentrations is also mentioned. These terms all need to be defined and used carefully, as they are not the same thing. This applies in the main body of the manuscript also. Second, more detail about the sites and methods are needed. For example, the response variables of the various tests are not given in the abstract. Readers will want to know what was actually measured (e.g. chlorophyll, biomass, etc) and the context.

Overall, the manuscript’s structure is too unorganized and sprawling. Move all the results into the Results section, broken into logical chunks. Then stick to the same structure in the Discussion. Some text could easily be removed without losing any important points. I have tried to point out some examples below.

Specific Suggestions

1) Line 11: Regardless of N concentration?

2) Line 12: Move “and by about…L P” to the end of the sentence.

3) Line 15: Proportions of nutrients taken up is meaningless without context.

4) Line 18: Across all three tests? Just ambient concentrations or amended ones too?

5) Line 32: The potential for N limitation in western US streams has been known for more than forty years, so P limitation is not assumed in the authors’ study region.

6) Line 35: “of which nutrient”

7) Lines 35-36: Here’s a definition of limitation, but it is theoretical, not operationally very useful (how does one measure supply?) and cannot be applied to the methods used here.

8) Line 48: But mostly it is because concentration is a poor indicator of supply rate.

9) Line 55: “The approach here…”

10) Lines 64-69: Convert to straight text.

11) Lines 70-96: None of this text belongs in the introduction. Move to the methods or discussion.

12) Lines 99-100: Duplication of “evaluate” and “evaluation.” Used again in the next sentence.

13) Line 154: “for a common”

14) Line 181: How is it subtly different?

15) Lines 191-194: What were the treatments?

16) Line 196: How were the plants subsamples for some of these responses (e.g. root length)?

17) Line 197: What is an “effect concentration”? What about the other responses? How were they analyzed?

18) Line 245: I am not sure that I buy that this step function is real.

19) Line 249: How was the periphyton sampled?

20) Line 281: Change “ruined” to “lost”

21) Line 282: “in all treatments”

22) Line 282-283: Avoid empty figure citations such as these. Cite the plot while describing it instead.

23) Line 302: These aren’t conclusions. Change to something like “Results from the NDS experiments…”

24) Line 319: An alternative conclusion is that strict co-limitation lends itself to control of just one of the nutrients.

25) Lines 334-350: Maybe this text on suppression could be moved to later in the discussion. It is a minor component of the results.

26) Line 358: I suggest that the authors move all these results to the results section. There is no real reason to wait until the discussion to present these data. It just makes for a longer and more confusing Discussion.

27) Line 366: Again, I question the validity of the step function.

28) Line 380: “Account for the large…”

29) Line 381-405: This section is long and does not add much to the manuscript. It’s hard to know what Lemna can tell us about macrophytes generally.

30) Line 397: Use another term instead of “shape shifters”

31) Line 409: “suggests why” (the subject is “magnitude”)

32) Line 415: Why can P concentrations be ignored?

33) Line 444: “allow” (the subject is “periods”)

34) Line 456: Again, this should be in the results.

35) Line 510: Use another term instead of “knobs”

36) The text’s font size switches between sentences in the manuscript, which is distracting to reviewers.

37) Phosphorus is often misspelled throughout the manuscript.

38) Throughout: N-limited, P-limited, community-level, low-nitrogen, nutrient-saturated, rate-limited when modifying a noun.

39) Fig. 4: Just because a logistic model fits these data does not mean that the fit is valid or biologically meaningful. I am not convinced.

40) Fig. 9: The legend states that there are top, middle and bottom plots.

6. PLOS authors have the option to publish the peer review history of their article (what does this mean?). If published, this will include your full peer review and any attached files.

Reviewer #1: No

Reviewer #2: No

---

## [Author Response · Author response to Decision Letter 0]

26 Mar 2021

[The responses to comments are uploaded in a separate file "Rebuttal letter.docx" that is easier for humans to read than the following plain text. Per request of the editorial office, the duplicative, plain text comments and responses follow.]

Editorial Comments

1. Please ensure that your manuscript meets PLOS ONE's style requirements,

We have reviewed these and believe it is consistent with style requirements. It was not clear whether this comment was a standard admonition or if there was something in the manuscript that stood out. Please advise if something needs correction in the revised version. 

2. In your Methods section, please provide additional location information of the study sites, including geographic coordinates for the data set if available.

Table 2 is in the methods section (list of study sites). The footnote refers readers to the SI for this information. Also in the footnote we have added reference to the data repository which also contains all geographic coordinates for the study sites.

3.We note that you have indicated that data from this study are available upon request. PLOS only allows data to be available upon request if there are legal or ethical restrictions on sharing data publicly.

In our original submittal, in the Editorial Manager query for ‘Data Availability’ we replied “Yes-all data are fully available without restriction” and we go on to describe the URLs for the public repositories from which the datasets may be downloaded.

Several instructions on how to present funder and acknowledgements were made, including: 

“We note that you have provided funding information that is not currently declared in your Funding Statement. However, funding information should not appear in the Acknowledgments section or other areas of your manuscript. We will only publish funding information present in the Funding Statement section of the online submission form. Please remove any funding-related text from the manuscript and let us know how you would like to update your Funding Statement.”

The financial information is removed from the acknowledgements and a revised funding disclosure has been provided in the cover letter as follows: 

“Funding for the experimental components of this project was provided by the U.S. Environmental Protection Agency’s Nutrient Criteria Program, through interagency agreement DW-14922442-01-0. Funding for the field component of this project was provided by the U.S. Geological Survey’s National Water Quality Assessment (NAWQA) program. No specific funding was received to write the article. The funders approved the general study design and encouraged publication of results but had no role in data collection and analysis, specific decisions to publish, or preparation of the manuscript. The manuscript was reviewed and approved for publication per U.S. Geological Survey Fundamental Science Practices.”

Additional Editor Comments:

I have now received review comments by two reviewers who agree on the strength of the paper and its contribution to nutrient management in streams and rivers. While both reviewers have provided specific and general comments on how the paper can be improved, reviewer 2 is very critical of the writing style used. Thus, the paper need reorganization using a parallel structure. There is also a need to shorten some sections of the paper as indicated.

We greatly appreciate the time and insights of the reviewers. We have considered and responded to all comments. Reviewer 1’s comments are in red text and Reviewer 2’s comments are in blue text.

Reviewer #1

This is a nice paper of practical and ecological importance. The growth assays combined with the field measures makes the work stronger and helps us start to understand linkages between laboratory and field experiments commonly used to assess nutrient limitation in streams. Nutrient pollution in streams is an important management issue, so the work has practical implications as well.

My comments are mainly on details, not the overall scope or conclusions of the manuscript.

Line 152. Was the algal culture axenic?

Yes, revised, and added source

Line 168. Given that multiple ANOVA tests were run, was there any effort to do Bonferonni correction or a giant combined test of results?

No family-wise comparisons of the ANOVA across streams were made. This is because we were not making any comparisons of limitation results across streams.

Line 189. What light intensity was used for the two types of bioassay experiments?

Added this detail to the text

Line 206. Were the NDS agar solutions mixed with phosphorus at the last minute upon cooling? Otherwise inhibitory compounds can be produced (e.g. reference 8). While later on the authors state that the agar was not autoclaved, this still could happen with boiling. It really is the best explanation (in my mind) for the inhibitory response.

The agar was off the hot plate and was starting to cool when the nutrients were added, but it couldn’t be allowed to cool too much, or else the nutrient additions wouldn’t mix fully and the agar has to be warm in order to pour it. Added this detail to methods. While we have no basis to refute Reviewer 1’s comment that bringing agar to a boil can introduce inhibitory compounds, we did not see anything within reference 8 or any other sources that could clarify this point or give us something to reference.

Line 233. A growth function as a function of nutrients is Monod, Michaelis-Menten is for nutrient uptake. Same mathematical form, but technically different. Line 264 makes that distinction.

Fair comment. The point of Figure 3 is to show that ambient waters with different P concentrations did indeed produce a concentration-dependent pattern in green algae growth, and that it reached an asymptote. We replaced the Michaelis-Menten curve fit in Figure 3 with an asymptotic fit, which shows the same thing, but avoids the conceptual problem that M-M and Monod are intended for rates. While rates could be calculated, the test design wasn’t optimal for it with green algae concentration as a starting and ending endpoint only.

Line 245. Stepped increase very interesting result

We thought so as well

Line 253. Larger plants have much more capacity to be plastic in their internal nutrient content and can store nutrient up from prior pulses. Line 261 supports this hypothesis. Lohman and Priscu (1992) show this for Cladophora (kind of on the border between a macro and a macrophyte…)

Lohman, K., and J. C. Priscu. 1992. Physiological indicators of nutrient deficiency in Cladophora (Chlorophyta) in the Clark Fork of the Columbia River, Montana. Journal of Phycology 28:443-448.

Lohman and Priscu (1992 is a good reference and we appreciate the reminder. Added to the references in line 265. However, the stored nutrients from prior pulses wouldn’t seem to be applicable here since by chance, we collected the stream sample for the N series at close to the highest N measured at any time during our 2-year sampling of the streams.

Line 456. Good explanation for Figures 8 and 9 that indicate that nutrient bioassays show that initial burst of growth, but are and are not subject to other controls of natural standing stock. Maybe consider that a quantile regression is good for maximum chlorophyll? This approach has been taken previously (or using maximum chl rather than seasonal means from standing stocks)

Yes, as noted below at line 469, this result is similar to what would likely be inferred from quantile regression. However, since we have direct experimental results, we were able to take the approach of overlaying the experimental results on the field collections.

Figure 2. I don’t think it is necessarily correct to calculate 95% confidence intervals when there is a significant interaction?

We think it is informative to compare the results of the single nutrient additions relative to the controls. The approach used here is conceptually similar to that used by Beck and Hall [42], although they used t-tests as opposed to confidence intervals. More generally, a variety of approaches have been reported in the literature for the interaction problem across the disciplines, and there is no single, most correct approach. Similar to the problems of nutrient addition in limnology, tests of equivalence in pharmacology to test Drug A against Drug B and against Drugs A&B together, vs. a placebo. In environmental toxicology, a similar problem arises with testing effects of pesticide A, B, and A+B, against controls. Many different data analysis approaches are used with these problems.

Figure 7 shows why longer term means are useful. For example figure a has a spike in TN and TP and a decline in benthic chlorophyll. This could be a sloughing event and the TN and TP made mostly of lost algae.

Concur, that is our interpretation as well - scour is the most likely cause for the decline in chlorophyll corresponding with spikes in TN and TP. Added the word 'scouring' at line 419, “likely attributable to scouring during high flows...” as shown in the interactive figure SI2. 

Figure 8 a. Not sure how you would get chlorophyll per kg of periphyton on the glass frits.

The glass frits serve as an artificial substrate and the analyses of the extracts give a mass and the glass frits have a known area

Reviewer #2

Reviewer 2 General Comments

This manuscript presents the results from a study that used multiple laboratory- and field-based approaches to explore the boundaries of N and P limitation in a group of western US streams that lie along a gradient of anthropogenic nutrient enrichment. Nutrient limitation was assessed using a combination of low-concentration threshold and high-concentration saturation responses with bottle bioassays of a single algal taxon, laboratory growth trials of Lemna and its epiphytes, and deployments of nutrient-diffusing substrates in the field. This is an interesting combination of approaches that could inform the setting of nutrient criteria in streams. Perhaps not surprisingly however, the manuscript struggles to organize all the various components of the results in a clear and logical way. I have both general and specific suggestions for its further improvement.

Overall, the manuscript’s structure is too unorganized and sprawling. Move all the results into the Results section, broken into logical chunks. Then stick to the same structure in the Discussion. Some text could easily be removed without losing any important points. I have tried to point out some examples below.

We are pleased Reviewer 2 found our manuscript of value and we appreciate the detailed review. We have considered the recommendations made about the writing style and we have made revisions.

General Comments

Abstract: The abstract suffers from a number of problems. First, the abstract seemingly uses the terms “limitation” and “saturation” interchangeably. Growth stimulation at low concentrations is also mentioned. These terms all need to be defined and used carefully, as they are not the same thing. This applies in the main body of the manuscript also. Second, more detail about the sites and methods are needed. For example, the response variables of the various tests are not given in the abstract. Readers will want to know what was actually measured (e.g. chlorophyll, biomass, etc) and the context.

Nutrient limitation and saturation are indeed very different, but the abstracts are not the place to expound on such details. If readers are (hopefully) interested in these sorts of details, they will have to read beyond the abstract. Since we realized we were 80 words over PLOS’s 300 word limit for the abstract, some words and details were removed.

Overall, the manuscript’s structure is too unorganized and sprawling. Move all the results into the Results section, broken into logical chunks. Then stick to the same structure in the Discussion. Some text could easily be removed without losing any important points. I have tried to point out some examples below.

Specific Suggestions

1) Line 11: Regardless of N concentration?

Yes, for the N concentrations tested. No room for additional detail in the abstract

2) Line 12: Move “and by about…L P” to the end of the sentence.

OK

3) Line 15: Proportions of nutrients taken up is meaningless without context.

Hopefully readers will be interested enough read beyond the abstract for context. Abstract word limits prevent adding additional detail

4) Line 18: Across all three tests? Just ambient concentrations or amended ones too?

Changed to “all tests and endpoints”

5) Line 32: The potential for N limitation in western US streams has been known for more than forty years, so P limitation is not assumed in the authors’ study region.

That is a good point. We amended the sentence so that it reads “this nutrient management presumption ...” to clarify that we are referring to management strategies, not scientific findings. We concur with the comment that the potential for N limitation has been well known among the stream ecology community, and we discuss that at the beginning of the discussion. However, this point in the introduction is about the presumption made in regulatory ‘TMDL’ nutrient reduction plans used in the US. This is intended to be a crossover paper with regulatory environmental managers among the targeted readers.

6) Line 35: “of which nutrient”

Corrected, thank you

7) Lines 35-36: Here’s a definition of limitation, but it is theoretical, not operationally very useful (how does one measure supply?) and cannot be applied to the methods used here.

This is a fair criticism. As recommended, we added to the sentence that, operationally, nutrient limitation is inferred by limitations in growth rates or standing crops of algae or aquatic plants. We also introduced the counter-condition, saturation, with a reference for more information.

8) Line 48: But mostly it is because concentration is a poor indicator of supply rate.

Revised to acknowledge that ambient concentrations may not reflect supply rates, and added a citation about luxury uptake of nutrients

9) Line 55: “The approach here…”

Corrected, thank you

10) Lines 64-69: Convert to straight text.

We believe this suggestion is to change the numbered list to an in-line list within a paragraph. We made that change

11) Lines 70-96: None of this text belongs in the introduction. Move to the methods or discussion.

We think the introduction is the best place to introduce the study approach and rationale in general terms, and to give context for the methods used. We considered this comment but did not follow this suggestion.

12) Lines 99-100: Duplication of “evaluate” and “evaluation.” Used again in the next sentence.

Changed second “evaluation” to “test”

13) Line 154: “for a common”

Corrected, thank you

14) Line 181: How is it subtly different?

The differences were described in the preceding sentences, but this particular statement probably just added confusion. Sentence deleted.

15) Lines 191-194: What were the treatments?

The treatments are listed in Table 3

16) Line 196: How were the plants subsamples for some of these responses (e.g. root length)?

Added that these measurements were made on 10 randomly selected plants per aquaria 

17) Line 197: What is an “effect concentration”? What about the other responses? How were they analyzed?

Changed to: ‘For endpoints which responded to treatments, effects concentrations associated with percentile increases were estimated...’

18) Line 245: I am not sure that I buy that this step function is real.

We make it clear at lines 366-370 that these responses were unexpected. However, the data are what they are, variability was low, and we suggest a plausible mechanism (epiphyte competition). We recognize that different readers may interpret the results differently. We make all the data easily available such that readers could make their own interpretations if they wish. Maybe our interpretation will stand the test of time, maybe not. We think this is how open science should work.

19) Line 249: How was the periphyton sampled?

Added sentence to methods near line 197 that periphyton was scraped from the aquaria walls.

20) Line 281: Change “ruined” to “lost”

The word “lost” would be the correct word if at the end of the experiment we couldn’t find the NDS vials - that is, they were missing. They were not missing. They were ruined.

21) Line 282: “in all treatments”

Corrected, thank you

22) Line 282-283: Avoid empty figure citations such as these. Cite the plot while describing it instead.

Reasonable point. Deleted this transitional sentence.

23) Line 302: These aren’t conclusions. Change to something like “Results from the NDS experiments…”

The table includes nutrient limitation conclusions from the statistical testing, and these conclusions are accompanied by co-occurring environmental conditions. We think the table title is accurate, descriptive, and concise.

24) Line 319: An alternative conclusion is that strict co-limitation lends itself to control of just one of the nutrients.

That’s a fair point, and we agree it is reasonable to mention it here. Two sentences were added with citations to two papers that we thought captured this debate well. However, we are keeping it limited, cognizant of the earlier complaint from Reviewer 2 of a “sprawling’ discussion.” 

25) Lines 334-350: Maybe this text on suppression could be moved to later in the discussion. It is a minor component of the results.

We concur that this is not a major component of the results, however, the limitation, saturation, and inhibition aspects of the NDS are all related and it flows better to keep them together. 

26) Line 358: I suggest that the authors move all these results to the results section. There is no real reason to wait until the discussion to present these data. It just makes for a longer and more confusing Discussion.

The N:P ratios are secondary interpretations from the experimental results, and pool results across all tests and endpoints. Thus, it seems to fit in the Discussion, and splitting it up would unnecessarily increase length. Just because a point of discussion is illustrated with a graph, it does not follow that the graph must appear in the Results section

27) Line 366: Again, I question the validity of the step function.

Please see the response to Reviewer 2’s comment #18

28) Line 380: “Account for the large…”

Corrected, thank you

29) Line 381-405: This section is long and does not add much to the manuscript. It’s hard to know what Lemna can tell us about macrophytes generally.

What Lemna can tell us (or not) about macrophytes, the dominant primary producers in some of our streams, generally is indeed the purpose of this section: What Lemna can tell us about macrophytes generally and the challenges of testing different macrophytes with different life histories. We posit that duckweed is likely to at least be relevant to the forms that do not have robust root structures

30) Line 397: Use another term instead of “shape shifters”

The ability of some taxa to shift their shapes under different flow conditions is truly remarkable, especially the Potamogetons. This seems a reasonable thing to mention.

31) Line 409: “suggests why” (the subject is “magnitude”)

Corrected, thank you

32) Line 415: Why can P concentrations be ignored?

We concur that this could be considered too conclusive. Toned down to “therefore, the N concentrations may be focused upon.”

33) Line 444: “allow” (the subject is “periods”)

Corrected, thank you

34) Line 456: Again, this should be in the results.

In our approach, the primary results presented in the Results section are the experimental results. Secondary analyses exploring implications or to help interpret independent field surveys are what we consider to be discussion.

35) Line 510: Use another term instead of “knobs”

OK. Changed to ‘functions’

36) The text’s font size switches between sentences in the manuscript, which is distracting to reviewers.

Thank you. We have given the text another going over and hopefully have caught all these.

37) Phosphorus is often misspelled throughout the manuscript.

Thank you. Three “phosphorous” instances were found and corrected to “phosphorus.”

38) Throughout: N-limited, P-limited, community-level, low-nitrogen, nutrient-saturated, rate-limited when modifying a noun.

We took another pass through the text looking for these instances. The ‘rules’ for hyphenating are a bit complicated, but we believe we have addressed any ambiguity.

39) Fig. 4: Just because a logistic model fits these data does not mean that the fit is valid or biologically meaningful. I am not convinced.

Please see the response to Reviewer 2’s comment #18

40) Fig. 9: The legend states that there are top, middle and bottom plots.

Thank you for catching those relicts of an earlier version.

---

## [Decision Letter · Decision Letter 1]

3 May 2021

PONE-D-20-39177R1

Nutrient limitation of algae and macrophytes in streams: integrating laboratory bioassays, field experiments, and field data

PLOS ONE

Dear Dr. Mebane,

Thank you for submitting your manuscript to PLOS ONE. After careful consideration, we feel that it has merit but does not fully meet PLOS ONE’s publication criteria as it currently stands. Therefore, we invite you to submit a revised version of the manuscript that addresses the points raised during the review process.

We look forward to receiving your revised manuscript.

Kind regards,

Frank Onderi Masese, Ph.D

Academic Editor

PLOS ONE

Journal Requirements:

Reviewers' comments:

Reviewer's Responses to Questions

**Comments to the Author**

1. If the authors have adequately addressed your comments raised in a previous round of review and you feel that this manuscript is now acceptable for publication, you may indicate that here to bypass the “Comments to the Author” section, enter your conflict of interest statement in the “Confidential to Editor” section, and submit your "Accept" recommendation.

Reviewer #2: (No Response)

2. Is the manuscript technically sound, and do the data support the conclusions?

Reviewer #2: Yes

3. Has the statistical analysis been performed appropriately and rigorously? 

Reviewer #2: Yes

4. Have the authors made all data underlying the findings in their manuscript fully available?

Reviewer #2: Yes

5. Is the manuscript presented in an intelligible fashion and written in standard English?

Reviewer #2: Yes

6. Review Comments to the Author

Reviewer #2: Review of PONE-D-20-39177R1 – “Nutrient limitation of algae and macrophytes in streams: integrating laboratory bioassays, field experiments, and field data” by Mebane, Ray and Marcarelli.

This manuscript presents the results from a study that used multiple laboratory- and field-based approaches to explore the boundaries of N and P limitation in a group of western US streams that lie along a gradient of anthropogenic nutrient enrichment. Nutrient limitation was assessed using a combination of low-concentration threshold and high-concentration saturation responses with bottle bioassays of a single algal taxon, laboratory growth trials of Lemna and its epiphytes, and deployments of nutrient-diffusing substrates in the field. I reviewed the original version of the manuscript. Overall, I find that the revision improved, even if the authors stuck to their guns regarding some of my criticism. I have both general and specific suggestions for its further improvement.

General Comments

More detail is needed about how the water samples were treated and analyzed. “Total” concentrations usually indicate that the sample is unfiltered and digested. Is this what the authors mean? If the authors actually mean “total dissolved” or “dissolved inorganic” then this should be made clear by laying it out in the methods and using the correct terminology.

Specific Suggestions

1) Line 3: “…with water collected from nine streams in an agricultural…”

2) Line 4: “alga”

3) Line 5: “…test of periphyton were conducted with nutrient-diffusing…”

4) Line 8: Periphyton? Do the authors mean epiphyton?

5) Line 11: “alga”

6) Line 14: Font size changes.

7) Line 19: “Our approach…”

8) Lines 16-18: Do the authors mean total dissolved concentrations here?

9) Line 44: Italicize r

10) Lines 80-84: If this paragraph is a justification for using Lemna then make it clearer. As is stands, this paragraph doesn’t make much sense. I still disagree with the authors about the need to put these methods-related sections in the introduction.

11) Line 117: “Streams are listed in order…”

12) Line 137: Missing period.

13) Line 149: “to the ambient water samples”?

14) Line 179: “algal species”

15) Line 198: Provide details of software.

16) How were these water samples filtered and analyzed? This is important in interpreting the results, as well as what the authors mean by “total” concentrations.

17) Line 235: What kind of function was fit to the curve?

18) Line 250: Insert comma after “regression”

19) Line 262 and 264: “was removed”

20) Line 329: “alga”

21) Line 332: “tests”

22) Line 375: Insert comma after “treatments”

23) Line 431: “instream”

24) Throughout: N-limited, P-limited, community-level, single-species, dual-nutrient, nutrient-saturated, nutrient-diffusing, rate-limited where modifying a noun.

7. PLOS authors have the option to publish the peer review history of their article (what does this mean?). If published, this will include your full peer review and any attached files.

Reviewer #2: No

---

## [Author Response · Author response to Decision Letter 1]

9 May 2021

We have responded to all comments raised in the review of Revision 1 and have attached a point-by-point accounting of these in the file "Responses to comments R2.docx"

---

## [Editor Report · Decision Letter 2]

25 May 2021

Nutrient limitation of algae and macrophytes in streams: integrating laboratory bioassays, field experiments, and field data

PONE-D-20-39177R2

Dear Dr. Mebane,

We’re pleased to inform you that your manuscript has been judged scientifically suitable for publication and will be formally accepted for publication once it meets all outstanding technical requirements.

Kind regards,

Frank Onderi Masese, Ph.D

Academic Editor

PLOS ONE
---

## [Editor Report · Acceptance letter]

10 Jun 2021

PONE-D-20-39177R2 

Nutrient limitation of algae and macrophytes in streams: integrating laboratory bioassays, field experiments, and field data 

Dear Dr. Mebane:

I'm pleased to inform you that your manuscript has been deemed suitable for publication in PLOS ONE. Congratulations! Your manuscript is now with our production department. 

Kind regards, 

on behalf of

Dr. Frank Onderi Masese 

Academic Editor

PLOS ONE